# Introducing a New Detailed Long-Term COSMO-CLM Hindcast for the Russian Arctic and the First Results of Its Evaluation

**Vladimir Platonov** [1,*] and **Mikhail Varentsov** [2,3,4,5]

1. Department of Meteorology and Climatology, Lomonosov Moscow State University, Leninskie Gory MGU, GSP-1, 1, 119991 Moscow, Russia
2. Research Computing Center, Lomonosov Moscow State University, Leninskie Gory 1/4, 119234 Moscow, Russia; mvar91@gmail.com
3. A.M. Obukhov Institute of Atmospheric Physics, 3 Pyzhyovskiy Pereulok, 119017 Moscow, Russia
4. Hydrometeorological Research Center of Russian Federation, 11-13 Predtechenskiy Pereulok, 123242 Moscow, Russia
5. Moscow Center for Fundamental and Applied Mathematics, GSP-1, Leninskie Gory, 119991 Moscow, Russia
* Correspondence: vplatonov86@gmail.com

**Abstract:** Diverse and severe weather conditions and rapid climate change rates in the Arctic emphasize the need for high-resolution climatic and environmental data that cannot be obtained from the scarce observational networks. This study presents a new detailed hydrometeorological dataset for the Russian Arctic region, obtained as a long-term hindcast with the nonhydrostatic atmospheric model COSMO-CLM for the 1980–2016 period. The modeling workflow, evaluation techniques, and preliminary analysis of the obtained dataset are discussed. The model domain included the Barents, Kara, and Laptev Seas with ≈12-km grid spacing. The optimal model setup was chosen based on preliminary simulations for several summer and winter periods with varied options, and included the usage of ERA-Interim reanalysis data as forcing data, the new model version 5.05 with so-called ICON-based physics, and a spectral nudging technique. The wind speed and temperature climatology in the new COSMO-CLM dataset closely agreed with the ERA-Interim reanalysis, but with detailed spatial patterns. The added value of the higher-resolution COSMO-CLM data with respect to the ERA-Interim was most pronounced for higher wind speeds during downslope windstorms with the influence of mountain ranges on the temperature patterns, including surface temperature inversions. The potential applications and plans of further product development are also discussed.

**Keywords:** COSMO-CLM; COSMO; regional climate modeling; Arctic climate; extreme climate statistics; verification; hindcast; long-term hydrometeorological dataset

## 1. Introduction

The Arctic region is the most sensitive to global climate changes. In particular, the temperature increase is most intense there [1–3]. The Arctic climate system has many complex feedback systems manifested in different atmospheric circulation features and diverse regional anomalies of opposite signs [1,4,5]. The "Arctic amplification" of global warming and the accompanying environmental changes draw more attention to the Arctic climate and extreme weather events. Arctic warming, occurring above the "global warming" signal, results from dynamic processes in the atmosphere providing a poleward heat advection [6–9].

Meso-γ and meso-β scale processes with typical sizes of a few dozen km [10] play a significant role in the Arctic climate system [9,11,12]. For example, large long-lasting polynyas, leads and puddle areas, ice hummocks, and cracks form greater heat fluxes compared to solid ice areas. This makes a significant total contribution to the regional

increase in the heat balance and net radiation at high latitudes [9,13]. Mesoscale processes also significantly contribute to many severe weather phenomena. The most striking examples are polar lows [14], downslope windstorms [15], mountain winds, tip jets [16], mesoscale convective precipitation, etc.

Severe events have devastating impacts on the coastal infrastructure, as well as shelf oil and gas production, leading to significant costly damage and to human casualties occasionally. At the same time, the Arctic region is one of the regions that is the most sparsely covered by ground observation networks in the world. Thus, there is very poor information regarding the spatial structure of such severe events. Despite the fact that satellites (e.g., SAR, Sentinel, AMSR-E, and QuikSCAT) now provide useful spatially resolving information for various meteorological parameters, such data are irregular in time and still do not reach the required level of reliability and detailing for the reconstruction of three-dimensional atmospheric dynamics [17,18].

In contrast to sparse and fragmented observations, atmospheric reanalysis products may provide long-term and uniform hydrometeorological data detailed in space and time. However, the state-of-the-art global reanalysis datasets, such as ERA-Interim [19], NCEP/NCAR [20], MERRA-2 [21], ERA-5 [22], and NCEP-CFSR [23], have too coarse a spatial resolution to reproduce many mesoscale processes and associated extreme events adequately. Regional Arctic datasets are worth highlighting, primarily Arctic System Reanalysis (ASR) v1 and v2 [24,25], which is the only current pan-Arctic regional reanalysis for relatively long-term timescales (2000–2016).

ASR was obtained by dynamical downscaling of ERA-Interim data using the polar version of the Weather Research and Forecast (WRF) regional atmospheric model [26] for a domain covering the whole Arctic at a 30 km grid in version 1 (version 2 has a 15 km grid spacing). Research showed that ASR reproduces polar lows more adequately compared to the global reanalyses [27]. However, even the 15 km grid spacing allows the reproduction of features with a horizontal scale of about 50 km and, more explicitly, excludes from analysis a broad spectrum of severe events of meso-$\gamma$ and particularly meso-$\beta$ scales.

Another important Arctic regional dataset is referred to as the Arctic CORDEX (Coordinated Regional Climate Downscaling Experiment) project [28–30]. This project embraces the downscaling of many regional atmospheric models using ERA-Interim reanalysis as forcing data for the 1979–2014 period, and CMIP5 model outputs for regional climate projections. In this way, the Arctic CORDEX results not in a reanalysis, but rather in a hindcast modeling. The regional simulations were downscaled to different horizontal model grid sizes: $\approx 0.4°$, $\approx 0.2°$, and $\approx 0.1°$. The COSMO-CLM model participated in the Arctic CORDEX project with simulation experiments for the wintertime periods [31] and for the Kara Sea region for 2002–2014 [32] with a $\approx 15$ km grid size.

Due to the increase in severe event repeatability [33,34], Arctic coastal development, and Northern Sea Route prospects, there is an emerging need for detailed long-term hydrometeorological and climatic information for the region with horizontal scales of at least several km. Regional atmospheric modeling is the most relevant tool to achieve this goal. Long-term model simulations could produce more reliable estimates of the current regional Arctic climate changes and extreme weather event frequencies.

Our work is devoted to creating such a long-term detailed dataset for the western Russian Arctic region using the COSMO-CLM regional atmospheric model. We created a detailed hindcast for a period of 37 years, from 1980 to 2016, with $\approx 12$ km horizontal grid spacing. The aim of this paper is to introduce the new Arctic COSMO-CLM hindcast to the scientific community and discuss the first results of its evaluation, primarily its added value over coarse reanalysis data used for the model forcing.

The structure of the paper is as follows. The next section describes the COSMO-CLM model, modeling framework, and domain as well as specific details of the model setup, determined from the set of preliminary simulations and their evaluation scheme of the final long-term simulations. The Results section presents a summary of long-term runs, characteristics of the dataset and its availability, and a comparison between the obtained

dataset and the ERA-Interim reanalysis. Potential dataset applications and the limitations of the study are considered in the Discussion section. The last section outlines the conclusions and plans for further research and development related to the presented dataset.

## 2. Data and Methods

### 2.1. Model Description

The COSMO-CLM (CCLM) model (ver. 5.x) was used as the main tool for the creation of the long-term meteorological dataset. CCLM is the climate version of the non-hydrostatic mesoscale atmospheric COSMO model, including various modifications and extensions adapted to the long-term numerical experiments. It was developed by the German Weather Service (DWD) and CLM-Community [35–37]. The model equations are solved on the rotational Arakawa C-grid [38] in latitude–longitude ($\lambda,\varphi$) coordinates with a pole tilt to minimize the issue of longitude convergence at the pole. The height coordinate is the terrain-following hybrid Gal-Chen coordinate $\mu$ ($\sigma$-z system) [39,40].

The standard configuration of the CCLM model includes the Runge–Kutta integration scheme with the fifth advection order. There is an option to apply the spectral nudging technique [41–43]. The Ritter and Geleyn radiation scheme [44] is based on the $\delta$ two-stream version of the radiation transfer equation. The moist and shallow convection is parametrized using Tiedtke mass-flux schemes with equilibrium closure based on moisture convergence [45].

Turbulence is described by a prognostic turbulence kinetic energy (TKE) based scheme, with a 2.5-order closure [46]. For the land grid cells, the TERRA [47,48] surface active layer model is used, and ocean and sea ice surface temperatures are taken from larger scale forcing data. Lake properties are treated in the model using the FLAKE parameterization [49]. A full description of the COSMO model physics, dynamics, and parameterizations is available elsewhere [50].

### 2.2. Experimental Design

#### 2.2.1. Model Setup

The model was used in our study for dynamical downscaling of the global reanalysis data for the model domain with $\approx$12 km ($\approx$0.108°) horizontal grid spacing. The reanalysis provided the forcing data for the model, i.e., it determined the model's initial and lateral boundary conditions, as well as the sea ice and sea surface temperatures. The domain covers most of the area of the Russian Arctic, including the Barents, Kara, and Laptev Seas (Figure 1).

The model grid was configured with 50 vertical levels, 11 of which were in the lowest 1 km, with the lowest model level at 10 m and an atmospheric column height of 22,000 m. The TERRA scheme was configured using nine soil layers with center depths of 0.005, 0.025, 0.07, 0.16, 0.34, 0.7, 1.42, 2.86, 5.74, and 11.5 m. External parameters describing the surface properties were obtained via the EXTPAR (External Parameters) tool, version 5.0 [51], from GLOBE (surface orography), MODIS (soil properties and albedo), and Globcover2009 [52] (vegetation cover, root depth, land fraction, etc.).

The final model configuration (selection of the model options and parameterizations) was obtained by running and evaluating several preliminary simulations with different model settings. These experiments were performed for two contrasting periods: summer (August–September 2015) and winter (December–January 2012–2013), allowing us to examine the model's capability to reproduce various atmospheric conditions.

Different model options were varied in 14 preliminary numerical experiments (Appendix A Table A1). First, we compared simulations forced with ERA-Interim (ERAI) [19] ($\approx$0.75$^0$ grid spacing, *_erai acronym in Table A1) or ERA-5 [22] ($\approx$0.3$^0$ grid spacing, *_era5 acronym) reanalyses. We also compared configurations with the spectral nudging technique switched on (*_sn acronym in Table A1) and off. This technique [41] suggested an assimilation of the large-scale atmospheric circulation from forcing data not only on the domain boundaries, but also inside the whole domain. This binds the large-scale components of the internal model

mode to the forcing data, and prevents the model from moving away from realistic large-scale patterns of atmospheric circulation. Spectral nudging was applied to wind and temperature fields above the 850 hPa level, with an assimilated wave scale of about 550 km. This length scale was selected based on recommendations from the literature [41–43,53–55] and the authors' previous experience [56,57] in order to bind a synoptic-scale circulation in the model to the reanalysis data but to avoid dumping the mesoscale processes.

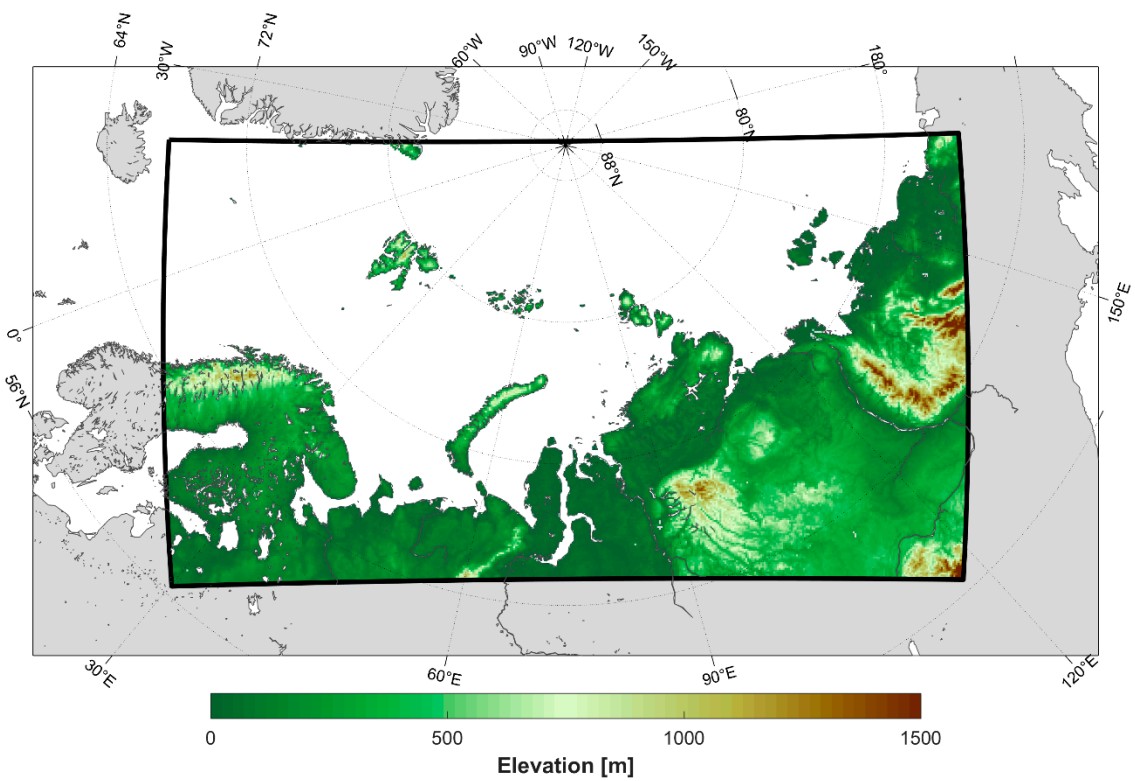

**Figure 1.** Scheme of the model domain with grid spacing ≈12 km, used for simulations in our study, where its borders are shown by a black rectangle, and the elevation data used in the model are shown.

We further compared two different model versions: 5.0 as the reference one and a newer 5.05 version (*_v505 runs). The latter allows application of the so-called ICON-based physical package, unified with the recent developments of the ICON model [58–60]. The new physical package includes a revised parameterization of the atmospheric boundary layer (ABL) turbulence, which improves the representation of the stably stratified ABLs. Excessively strong ABL mixing in stably stratified conditions and the resulting warm bias for the 2-m temperature were known shortcomings of the older turbulence scheme in COSMO, whereas application of the new physical package mitigates these problems [58–62].

Hence, we used the 5.05 version with activated ICON-based physics, as suggested in [63]. For the 5.0 model version, we attempted to reach a similar effect by changing the constants, which affects the vertical turbulent diffusion parametrization (*_turb). The minimal value of the turbulence drag coefficient was reduced by four times compared to the reference value, and the subgrid scale of temperature heterogeneity affecting the TKE generation was reduced by five times. Previous studies found that such changes significantly improved the simulation results for the cases with stable ABL stratification [64,65]. Finally, we investigated the model sensitivity to the spin up length in runs without any spin up and with 1-month spin up (*_long runs).

Verification of the preliminary simulations was based on the 2-m temperature (T_2M), 10-m wind speed (V_10M), and sea level pressure (PMSL) data from 466 meteorological stations covered by the model domain and located to the north from 60° N (Figure 2). Model grids for comparison were defined for each station based on the least root mean square error

(RMSE) between the four nearest grid points. The main statistical metrics of verification for each variable were the mean error (bias), RMSE, and correlation coefficient (R). These estimates were carried out for all stations and for the coastal and inland stations separately.

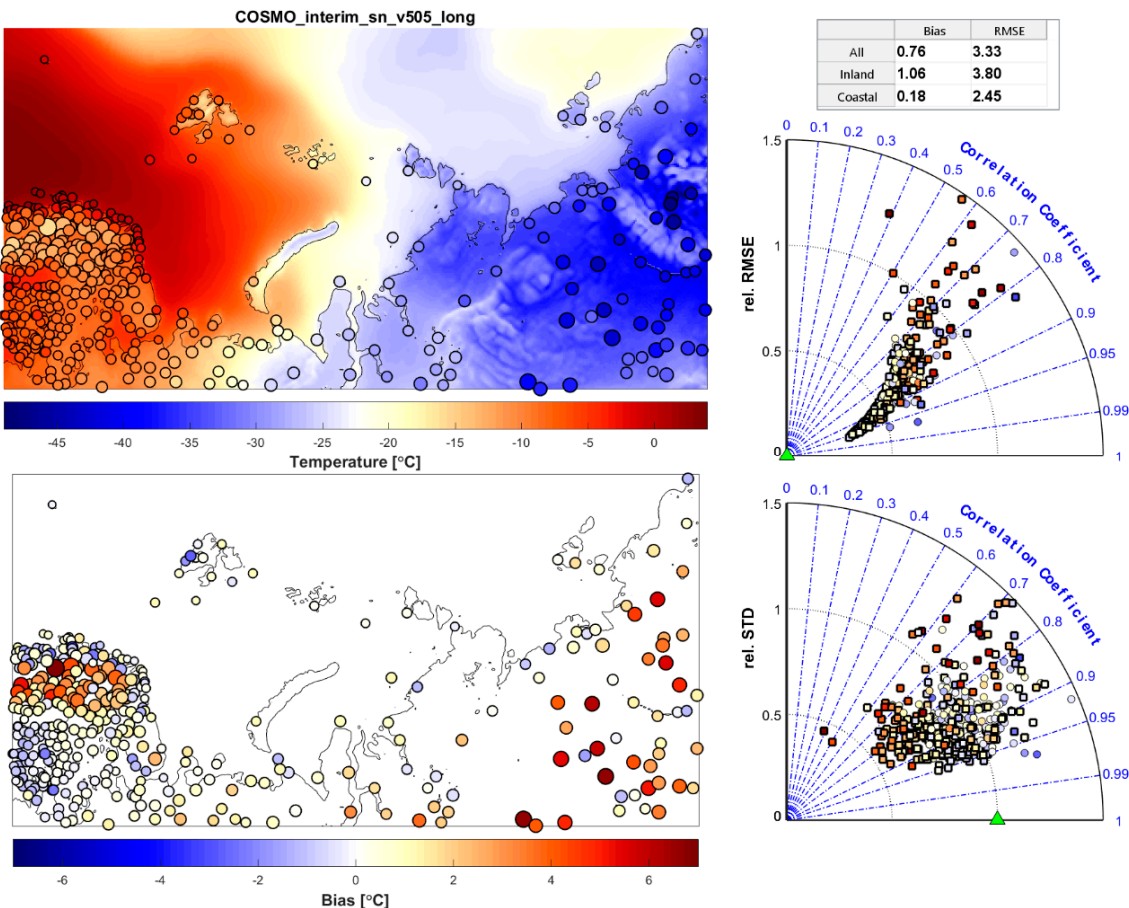

**Figure 2.** Example of the verification plot for the 2 m temperature January 2013 "COSMO_erai_sn_v505_long" experiment (see Table A1 for acronym explanation). A map of the modeled (color background) and observed (round markers) mean monthly temperature is shown on the top left. The marker size is proportional to the RMSE value for the given station. The map of mean errors is shown on the bottom left. Modified Taylor diagrams show the correlation coefficient and RMSE values normalized on the observed standard deviation for each station (top right); the correlation coefficients and ratio between the modeled and observed standard deviation for each station (bottom right). Green triangles show an ideal accordance case. The table on the top right shows the mean statistics for a given experiment over the domain: bias and RMSE for all and the inland and coastal stations.

The results of the preliminary simulations are briefly described below with the major verification scores presented in Table 1, and a more detailed description may be found in [66]. An example of a verification plot for a 2 m temperature in the simulation run with spectral nudging and the 5.05 model version for winter is given in Figure 2. Similar plots were prepared for all listed simulations, periods, and variables (90 figures in total), and analyzed together with the formal scores when selecting the best model configuration.

The reference model (version 5.0) without spectral nudging demonstrated the worst results with a temperature bias of about 1 °C and an RMSE of about 3.5–4 °C. In particular, there were large errors over inland Eastern Siberia, where the model strongly underestimated the winter boundary layer freezing. This is a known problem of stable ABL reproduction in mesoscale models, including COSMO [64,65,67,68].

**Table 1.** Major evaluation scores for the preliminary simulations, root mean square error (RMSE), and correlation coefficient (R).

| Model Set Up | December–January 2012–2013 | | | | | | August–September 2015 | | | | | |
| | T_2M | | V_10M | | PMSL | | T_2M | | V_10M | | PMSL | |
| | RMSE | R | RMSE | R | RMSE | R | RMSE | R | RMSE | R | RMSE | R |
|---|---|---|---|---|---|---|---|---|---|---|---|---|
| COSMO_erai | 4.22 | 0.76 | 2.30 | 0.55 | 2.98 | 0.96 | 2.38 | 0.77 | 2.02 | 0.65 | 1.87 | 0.99 |
| COSMO_era5 | 4.19 | 0.76 | 2.30 | 0.57 | 2.77 | 0.97 | 2.34 | 0.79 | 2.00 | 0.67 | 1.70 | 0.99 |
| COSMO_erai_sn | 3.69 | 0.83 | 2.12 | 0.65 | 2.01 | 0.99 | 2.89 | 0.79 | 1.89 | 0.70 | 1.53 | 1.00 |
| COSMO_era5_sn | 3.70 | 0.83 | 2.10 | 0.66 | 2.13 | 0.99 | 2.29 | 0.81 | 1.87 | 0.71 | 1.42 | 1.00 |
| COSMO_erai_turb_sn | 3.38 | 0.84 | 2.12 | 0.65 | 2.08 | 0.99 | 2.35 | 0.79 | 1.89 | 0.69 | 1.57 | 1.00 |
| COSMO_era5_turb_sn | 3.37 | 0.85 | 2.09 | 0.66 | 2.18 | 0.99 | 2.35 | 0.81 | 1.88 | 0.70 | 1.45 | 1.00 |
| COSMO_erai_sn_v505 | 3.34 | 0.85 | 2.22 | 0.65 | 1.69 | 0.99 | 2.10 | 0.81 | 1.97 | 0.70 | 1.40 | 1.00 |
| COSMO_era5_sn_v505 | 3.33 | 0.85 | 2.24 | 0.67 | 1.63 | 0.99 | 2.16 | 0.82 | 1.97 | 0.70 | 1.34 | 1.00 |

The use of spectral nudging significantly reduced the model biases for temperature as well as for wind speed during both periods. Modification of the tuning parameters affecting vertical turbulence diffusion significantly decreased the RMSE of the winter temperature for inland and mountainous areas (up to 3.3 °C for all stations and 3.7 °C for inlands) and obtained slightly negative biases (approx. −0.5 °C).

The model sensitivity to turbulence scheme parameters was small for the coastal stations, for the summer period, and for the wind speed estimates. Almost similar effects for winter temperatures were reached with the newer model version 5.05 with an ICON-based physical package. Additionally, the new model version performed better than the 5.0 in summer. Experiments with a 1-month spin up demonstrated minor oppositely directed differences: slightly increased temperature biases for the winter, and vice versa for the summer.

Surprisingly, the use of newer and more detailed ERA5 forcing data did not demonstrate any clear advantage over the ERAI data in the paired experiments, at least for the temperature and wind speed. A weak improvement with the use of ERA5 forcing was found only for the pressure RMSE. Taking into account the much higher storage and processing time for ERA5 data, we considered the ERAI forcing to be a cost-effective solution for long-term simulations.

Based on the evaluation results, for our final long-term simulations, we used the 5.05 model version with the spectral nudging technique and ERAI forcing. This is the first investigation of model parameters sensitivity for such a large Arctic region.

### 2.2.2. Final Long-Term Experiments Scheme

For the final long-term simulations, a computational scheme with monthly reinitialized soil properties was suggested. Long-term regional climate simulations could be accompanied by systematic error accumulation associated with incorrect adaptations of the long-term model's soil property variations in fast-changing atmospheric conditions [69,70]. The possible reason for such biases is considered to be soil draining in the model, i.e., the thermal and moisture properties of rather deep soil layers.

Our approach suggested that the model was initialized on the first day of every month from a combined data file that merged all atmospheric and surface variables from the last model output file and the main deep soil variables from the ERAI reanalysis. The first and second upper soil layers (0.005 and 0.025 m) were also taken from the last model output as the most sensitive to fast soil–atmosphere interactions.

The temperature and water content on the rest soil layers (from the third to the last—the ninth) were taken from the reanalysis interpolated onto the model grid. In this scheme, we proposed the ERAI soil data as more objective and the reference. This joined file was shaped and used as the initial forcing for the following month's run. After the end of each run, the last monthly model output file was used to create the next initial file using reanalysis data. Hence,

the soil temperature and humidity were re-initialized every month; however, the atmospheric dynamics were kept almost undisturbed.

The suggested scheme could introduce biases and artificial noise to the model by changing the initial data each month, because the simulated deep soil conditions were slightly different than the corresponding reanalysis data. To check this issue, additional test simulations were carried out for four periods: June–July 2010, 2013 and November–December 2010, 2013 using 1 month as the spin-up period. These periods are referred to as contrast seasons and are characterized by significantly different background soil properties.

For each period, three model runs were evaluated: continuous runs for 3 months, runs with the suggested reinitialization scheme for 3 months, and longer continuous runs for 1 year (initialized on 1 January). Based on the model verification, according to the abovementioned technique, we did not find any significant differences between these runs in terms of the model errors for all periods. Hence, the suggested reinitialization scheme does not worsen the simulation results and can be applied for long-term simulations.

At the same time, this scheme did not bring any notable improvements, which possibly indicates a minor role of the soil drainage properties over the Arctic region in the long-term model simulations. This fact could be explained by the permanent high-level moistening conditions in most of the Arctic region; therefore, there were no significant soil dryness effects during drought periods, which was possibly the main source of the model error increase. Finally, the suggested reinitialization scheme was applied in the long-term simulations.

## 3. Results

### 3.1. Long-Term Dataset

The modeling technique described above was used to create the long-term meteorological hindcast over the model domain for the 1980–2016 period. Simulations were performed using the shared research facilities of the high-performance computing resources at Lomonosov Moscow State University, supercomputer "Lomonosov-2" [71]. The computations were very resource-intensive and expensive. We performed 444 monthly runs in total for the 1980–2016 period. Each run used 144 nodes, and more than 250,000 node-hours were used in total. The stored model output included more than 100 different hydrometeorological variables at the surface, as in the model levels within the atmosphere and soil (see Appendix B Table A2), with a total output volume of 120 Tb.

Sharing such a large volume of data online is a challenging task, and it is therefore not completed yet. However, we have shared a limited-volume dataset that includes the most important surface variables, namely the PMSL (mean sea level pressure, Pa), QV_2M (2 m specific humidity, kg/kg), T_2M (2 m temperature, K), TOT_PREC (total precipitation, kg/m$^2$), U_10M and V_10M (zonal and meridional wind speed, m/s), and VABSMX_10M (maximum 10 m wind speed without gust, m/s). This dataset is available at the Figshare repository [72].

The data for each year are stored in a separate subfolder, where each file contains the data for one variable for one year in 3 h intervals. In this way, each variable has a three-dimensional grid, including 226 latitude grids, 421 longitude grids, and 2928 (or 2936 for leap years) time steps. All datafiles are presented in the NetCDF 4 classic format, and chunked and compressed to the fourth level using the *nccopy* utility according to the CF 1.4 conventions. There are also two additional NetCDF files, the first one containing latitude and longitude coordinate variables, the second one containing the time variable. A link to the open-access dataset is given in Appendix C. We plan to publish the full dataset later, but, for now, the data that are not presented in [72] may be accessed by request to the corresponding author.

The obtained amount of data raises technical challenges not only for sharing, but also for analysis and processing. Therefore, in this paper, we demonstrate only the preliminary results of the data analysis for the surface wind speed and temperature climatology for the two periods, 1980–1990 and 2010–2016.

### 3.2. Added Value of the COSMO-CLM Hindcast

As a first stage of the evaluation of the new CCLM Russian Arctic dataset (≈12 km grid), it was compared with the forcing data, i.e., ERAI reanalysis (0.75° horizontal grid) to investigate the added value of the higher-resolution modeling. The surface wind and temperature fields were assessed for the 1980–1990 and 2010–2016 periods, including the average 10 m wind speed, frequencies of extreme wind speeds above the 17.2 and 20.8 m/s (corresponding to the "gale" and "strong gale" of 8 and 9 Beaufort numbers) thresholds, as well as the 1% and 99% percentiles and average values of the 2 m temperature according to ERAI and CCLM datasets. We excluded from the analysis the so-called relaxation zone of 10 grids from each side of the domain, where the internal model solution is nudged against the forcing data [73,74].

Further results are presented as differences between the CCLM and ERAI data interpolated onto the CCLM grid to demonstrate the added values of the higher-resolution model to the regional climatology of the analyzed variables, which is a common method for regional climate modeling studies [75–78].

The comparison between the mean 10 m wind speed for the 2010–2016 period (Figure 3a,b) showed generally good and unbiased agreement between the two datasets. The mean CCLM-ERAI difference was about 0.1 m/s over the whole domain, 0.4 m/s over the land grids, and −0.1 m/s over the sea grids, without a clear difference between the summer and winter seasons (Table 2). However, there were many significant regional variations caused by the more detailed description of the surface properties and mesoscale atmospheric dynamics with the CCLM model (Figure 3c,d). Added values manifested in the significant wind speed increase over the Arctic islands, including Novaya Zemlya, Svalbard, and Severnaya Zemlya, which are due to the well-known mesoscale local wind speed accelerations due to downslope windstorms, such as bora [15,79,80].

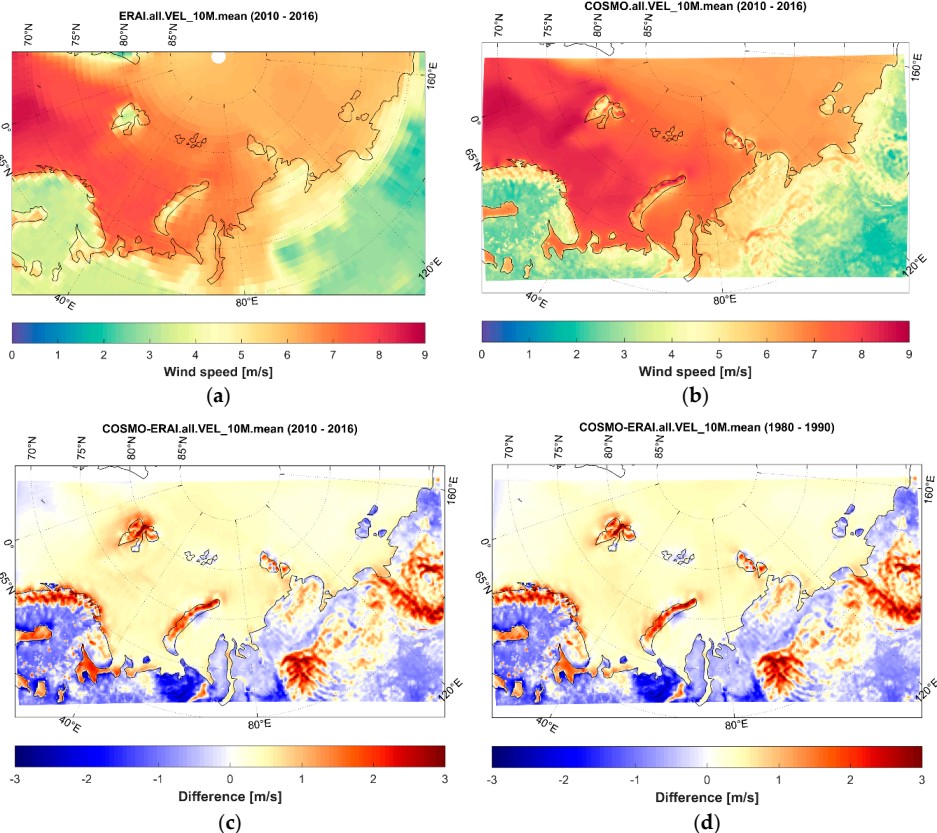

**Figure 3.** Average 10 m wind speed (m/s) according to the ERA-Interim reanalysis (**a**) and COSMO-CLM hindcast (**b**) for the 2010–2016 period. COSMO-CLM and ERA-Interim differences are presented for the 2010–2016 (**c**) and 1980–1990 (**d**) periods.

**Table 2.** The mean and percentile values of the differences between COSMO-CLM and ERA-Interim data for different variables and periods. Averaging was done for the whole year (if not specified), for the winter (DJF), and for the summer (JJA) months.

| Variable, Period | Mean Difference, Whole Domain | Mean Difference, Land Grids | Mean Difference, Sea Grids | First Percentile of Difference, Whole Domain | 99th Percentile of Difference, Whole Domain |
|---|---|---|---|---|---|
| VEL_10M, 1980–1990 | 0.13 | 0.34 | −0.14 | −1.63 | 2.22 |
| VEL_10M, 2010–2016 | 0.14 | 0.37 | −0.13 | −1.63 | 2.19 |
| VEL_10M, 1980–1990 (DJF) | 0.03 | 0.37 | −0.38 | −2.25 | 2.47 |
| VEL_10M, 2010–2016 (DJF) | 0.06 | 0.42 | −0.38 | −2.14 | 2.47 |
| VEL_10M, 1980–1990 (JJA) | 0.23 | 0.24 | 0.23 | −1.06 | 2.15 |
| VEL_10M, 2010–2016 (JJA) | 0.25 | 0.24 | 0.26 | −1.12 | 2.14 |
| T_2M, 1980–1990 | −0.03 | 0.10 | −0.19 | −1.67 | 1.72 |
| T_2M, 2010–2016 | −0.24 | 0.01 | −0.56 | −1.87 | 1.39 |
| T_2M, 1980–1990 (DJF) | 0.30 | 0.16 | 0.48 | −2.18 | 4.81 |
| T_2M, 2010–2016 (DJF) | 0.04 | 0.08 | 0.00 | −2.51 | 4.85 |
| T_2M, 1980–1990 (JJA) | −0.37 | −0.05 | −0.76 | −2.70 | 1.31 |
| T_2M, 2010–2016 (JJA) | −0.42 | −0.07 | −0.85 | −2.61 | 1.14 |

Other important features were the wind speed increases over large lakes (e.g., Ladoga and Onega), in Tiksi Bay, fractionally in Southern Taymyr, the Putorana Plateau, and in mountainous areas of Eastern Siberia. These features suggest that the CCLM model reproduced large-scale circulation properties adequately; moreover, it captured many regional mesoscale patterns linked to the more detailed surface and relief description. Despite ongoing Arctic climate change, the model's added value for the wind speed was similar for the 1980–1990 and 2010–2016 periods.

The added value of the CCLM hindcast over ERAI was clearly expressed for wind speed frequencies above the 20.8 m/s threshold. The model resolved the areas with the highest probability of extreme wind speed over Svalbard, Severnaya Zemlya, Putorana Plateau, and Tiksi Bay, with the most striking example over Novaya Zemlya Island (Figure 4a,b). High wind speeds in these areas were associated with orography and remain unresolved by reanalysis data. Additionally, CCLM reproduced the higher repeatability of such strong winds over the Barents and Norwegian Seas.

The latter feature was particularly expressed in the winter season (Figure 4d) and may be associated with better-resolved polar lows, which are frequently observed over this region [14]. Patterns for the 17.2 m/s threshold were quite similar (not shown), with the highest added value of the CCLM dataset located in the same areas, and with an even clearer increase in the strong wind frequency over the Barents and Norwegian Seas. As for the wind speed, the difference in the model's added value for the extreme wind speed repeatability for the two periods, 2010–2016 and 1980–1990, was generally similar, with only minor differences (details not shown).

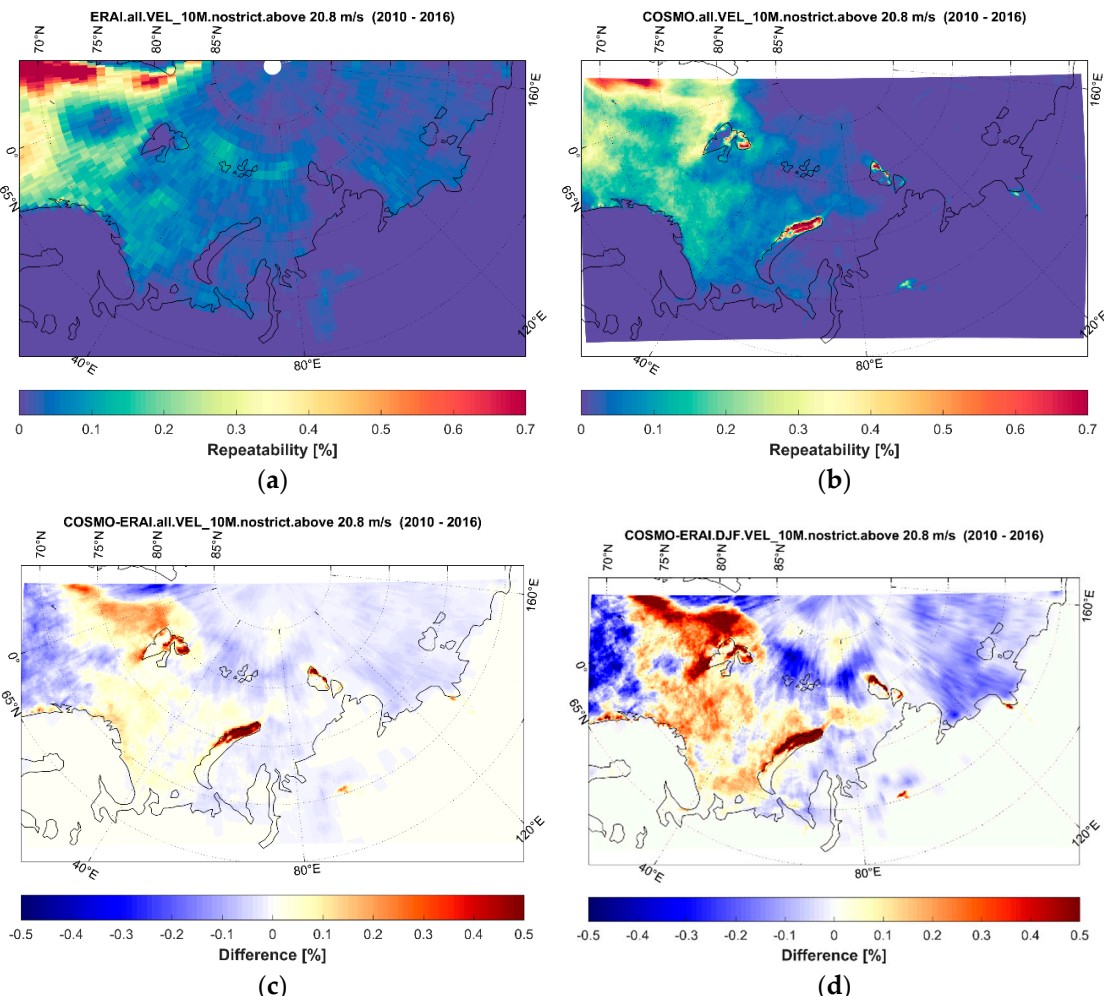

**Figure 4.** Frequency (%) of the 10 m wind speed above 20.8 m/s according to ERA-Interim reanalysis (**a**) and COSMO-CLM hindcast (**b**) for the 2010–2016 period. COSMO-CLM and ERA-Interim differences are presented for 2010–2016 over all months (**c**) and for the winter season (December–February) (**d**).

Analysis of the average 2 m temperature for the 2010–2016 period (Figure 5a,b) again indicated good consistency of the ERAI and CCLM datasets. On average, the CCLM hindcast had a slightly cold bias with respect to ERAI, with a mean difference between the two datasets of about −0.2 °C and −0.6 °C for land grids over the 1980–1990 and 2010–2016 periods, respectively, and a near-zero difference over the sea grids. The cold bias was expressed more strongly in the summer (Table 2). Despite the CCLM-ERAI biases being unsteady between the two periods, this difference was an order of magnitude lower than the pronounced temperature difference between these two periods, which reflects the well-known rapid climate warming occurring in the Arctic. The domain-average temperature difference between the two periods was 1.8 °C for CCLM and 2.0 °C for ERAI, with a warming hotspot of more than 5 °C to the north from Novaya Zemlya (Figure 5c). The spatial patterns of the temperature change for CCLM were almost similar to those in ERAI (not shown); however, the CCLM hindcast resolved the finer details, e.g., the difference in the warming rates for land and water grids, or for grids with different elevations.

The remarkable added value of the CCLM hindcast against ERAI was particularly evident for areas with complex terrain or with an abundance of lakes, e.g., over Scandinavia, Eastern Siberia, the Taymyr highlands, and the Novaya Zemlya ranges (Figure 5d). In general, the CCLM hindcast provided lower temperatures over plain areas and higher temperatures over mountain ranges. The latter was expressed in particular for the winter

season (Figure 5e), when the difference in mean temperature exceeded 5 °C for elevated areas in East Siberia.

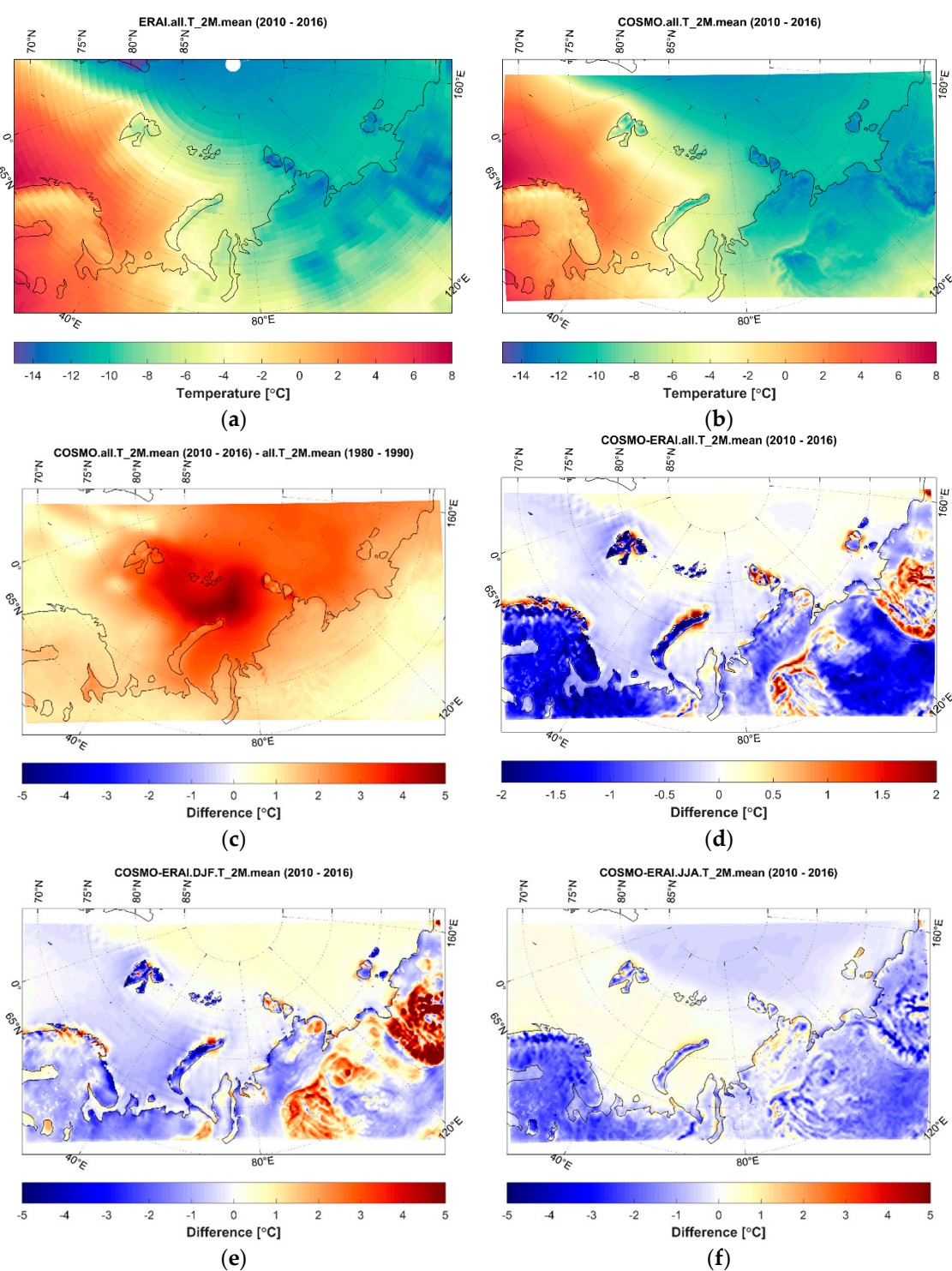

**Figure 5.** The average annual 2 m temperature according to the ERA-Interim reanalysis (**a**) and COSMO-CLM hindcast (**b**) for 2010–2016 period, the difference between the 1980–1990 and 2010–2016 periods for the COSMO-CLM hindcast (**c**), and the difference between COSMO-CLM and ERA-Interim temperatures for the 2010–2016 period for mean annual temperatures (**d**), for winter (DJF) temperatures (**e**), and for summer (JJA) temperatures (**f**).

At the same time, narrow valleys and lowlands in winter experienced lower temperatures in the CCLM hindcast. These features may be explained by better-resolved

temperature inversions in the CCLM hindcast, since they are predominant in the continental Arctic in winter [81]. On the contrary, the CCLM-ERAI temperature difference was strongly negative over the mountains in summer, reaching −4 °C (Figure 5f).

The revealed added value of the CCLM hindcast was even stronger if considering extreme temperature characteristics, e.g., the 1% temperature percentile for the 2010–2016 period (Figure 6). Whereas the mean CCLM-ERAI difference was close to zero, the higher-resolution hindcast strongly increased the low-temperature extremes for elevated mountain areas and decreased them in narrow valleys and over plains, with the temperature difference reaching 7 °C on both sides. As suggested above, such features indicate a better representation of the atmospheric stratification and strong wintertime temperature inversions, especially over complex terrain. The added value was also prominent over the Onega and Ladoga lakes and Western Kara Sea. The spatial patterns of the CCLM-ERAI difference were almost steady between the two periods considered.

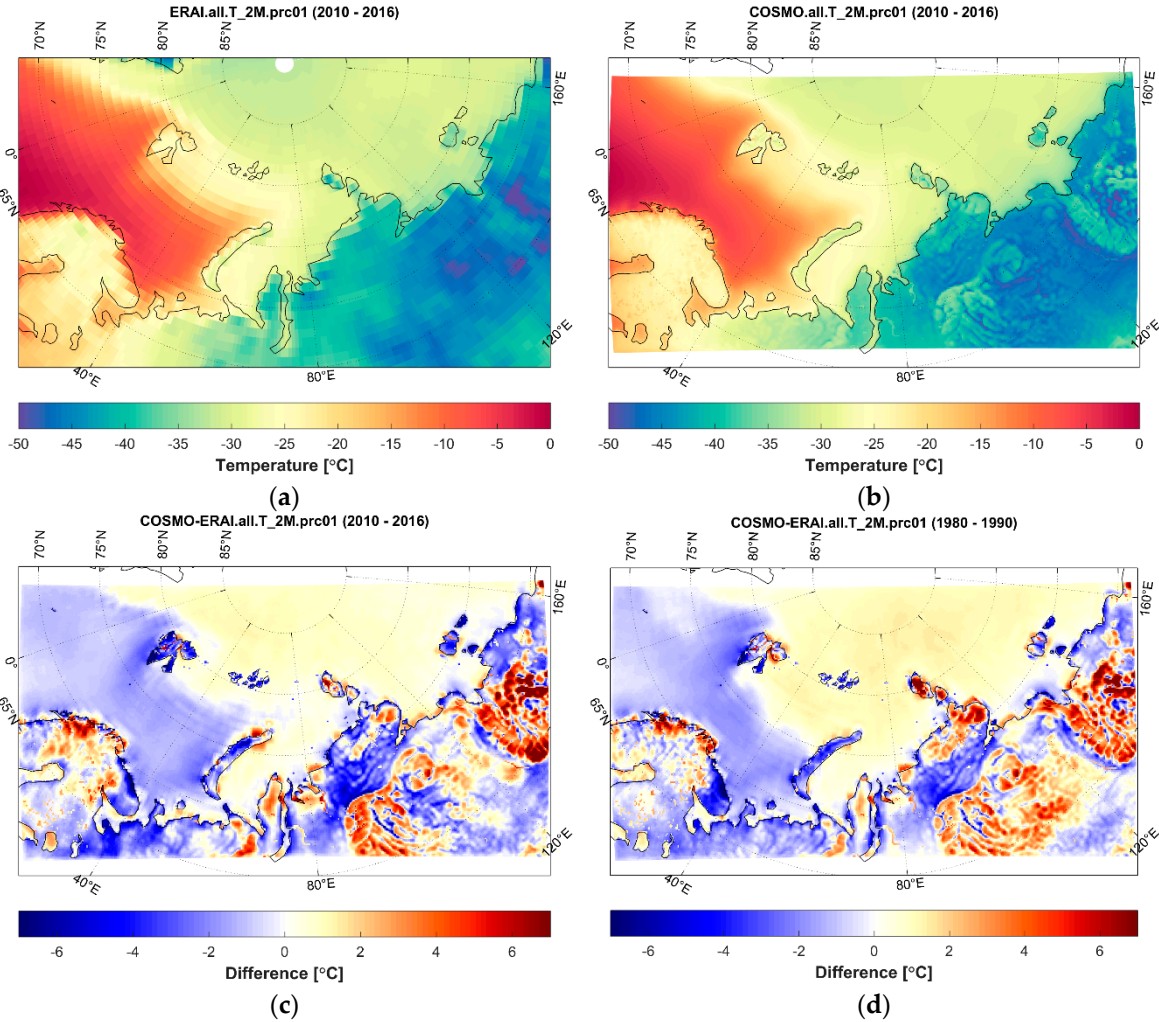

**Figure 6.** The 2 m temperature 1% percentile (°C) according to ERA-Interim reanalysis (**a**) and COSMO-CLM hindcast (**b**) for the 2010–2016 period. The COSMO-CLM and ERA-Interim differences are presented for the 2010–2016 (**c**) and 1980–1990 (**d**) periods.

## 4. Discussion

The first sight of the CCLM dataset created for the Russian Arctic demonstrated its large-scale agreement with the ERAI reanalysis in terms of the large-scale wind speed and temperature patterns. The higher-resolution CCLM dataset with 0.108° grid spacing revealed many mesoscale features that were not represented in the global reanalysis. These

included higher wind speed frequencies over the known downslope windstorm regions, orography, and the mountain range impact on the temperature patterns, including a better permission of near-surface inversions.

These added values are also associated with more detailed descriptions of the surface properties, coastline, and relief in the mesoscale model, as well as with a more accurate representation of the atmospheric dynamics due to the nonhydrostatic approach and more detailed physical parameterizations. This allows us to accept that the CCLM Russian Arctic hindcast provided physically justified hydrometeorological information regarding the Arctic region. At the same time, there are some objective limitations and restrictions, which need to be analyzed further.

There are, for example, sea ice conditions assimilated from global forcing without any dynamic approach, which could affect the local errors in the surface fluxes and the temperature in different sea ice edge regions or coasts. The absence of additional data assimilation from the stations and satellites in CCLM simulations could affect the errors in individual cases, including extreme events. Additionally, the soil temperature and moisture values must be compared and verified to investigate the effect of the implemented reinitialization scheme on the soil and surface flux modeling results.

There are many potential applications of the CCLM Russian Arctic hindcast. This dataset could provide more detailed estimates of the current regional climate and environmental changes, as well as extreme hydrometeorological events, their frequency, and trends. The hindcast can be used as forcing data to simulate the ocean's dynamics and wind waves [80,82], or for nested simulations with an atmospheric model for specific case studies (hazardous weather events, etc.). Applications of the CCLM hindcast include foreseeing studies devoted to coastal ecosystems, air–sea interactions (turbulent heat and moisture fluxes, as well as greenhouse gases), the climatology of extreme events, and many other fields.

The use of the new dataset appears especially promising for studying the climatology of mesoscale weather phenomena and, in particular, polar lows (PLs). The latter topic is experiencing significant public and scientific attention due to the challenges of the predictability of PLs and in understanding their climatology [83,84], but they are still poorly represented in coarse reanalysis data [14,85,86].

Figure 7 illustrates the difference between the new CCLM hindcast and the coarser ERAI and ERA5 reanalysis data with two example cases with PL activity in the Norwegian and Barents Seas, described in the literature for 30 March, 2013 [27], and for 13 December 2015 [87]. In both cases, two or three small but intense PLs appeared to the north of the Scandinavian peninsula at the border between the Norwegian and Barents Seas. These PLs can be clearly seen in the wind speed data obtained from an AMSR-2 radiometer on the AQUA satellite (data were obtained from Physical Oceanography Distributed Active Archive Center, https://podaac.jpl.nasa.gov/, accessed on 6 March 2021).

Figure 7a shows both PLs observed on 30 March, 2013, and Figure 7b shows two of the three PLs documented for 13 December, 2015. CCLM did not exactly capture the dynamics of the observed mesoscale eddies but, nevertheless, simulated similar PL activity, with wind speeds exceeding 20 m/s in this region (Figure 7c,d). For the first case, the model reasonably captured the eastern PL and, for the western one, simulated PL-like activity to the north from its actual location. For the second case, the model again nicely captured the eastern PL, and reproduced the western one with a slight delay in the dynamics and an underestimation of the wind speed.

Not surprisingly, the coarser ERAI data did not resolve any of these PLs (Figure 7e,f). The more recent and detailed ERA5 reanalysis also failed to reproduce the PL activity for the first case, and strongly underestimated the corresponding wind extremes in the second case (Figure 7g,h). The presented results are in line with other recent studies reporting the ability of the COSMO model to represent PLs over the Arctic seas [27,88], with a revealed increase in extreme wind repeatability over the Norwegian and Barents Seas in the CCLM hindcast against the ERAI data (Figure 4).

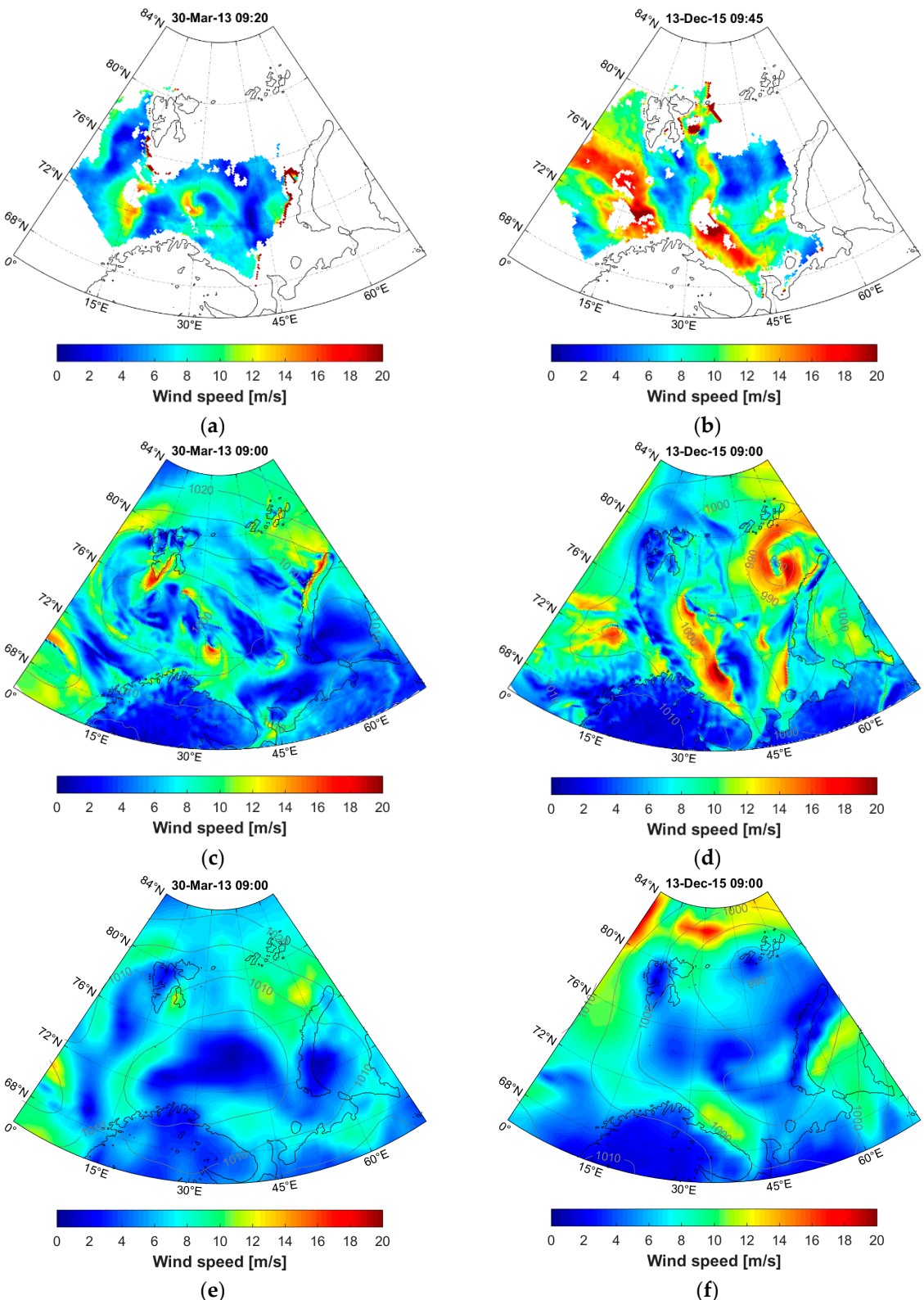

**Figure 7.** *Cont.*

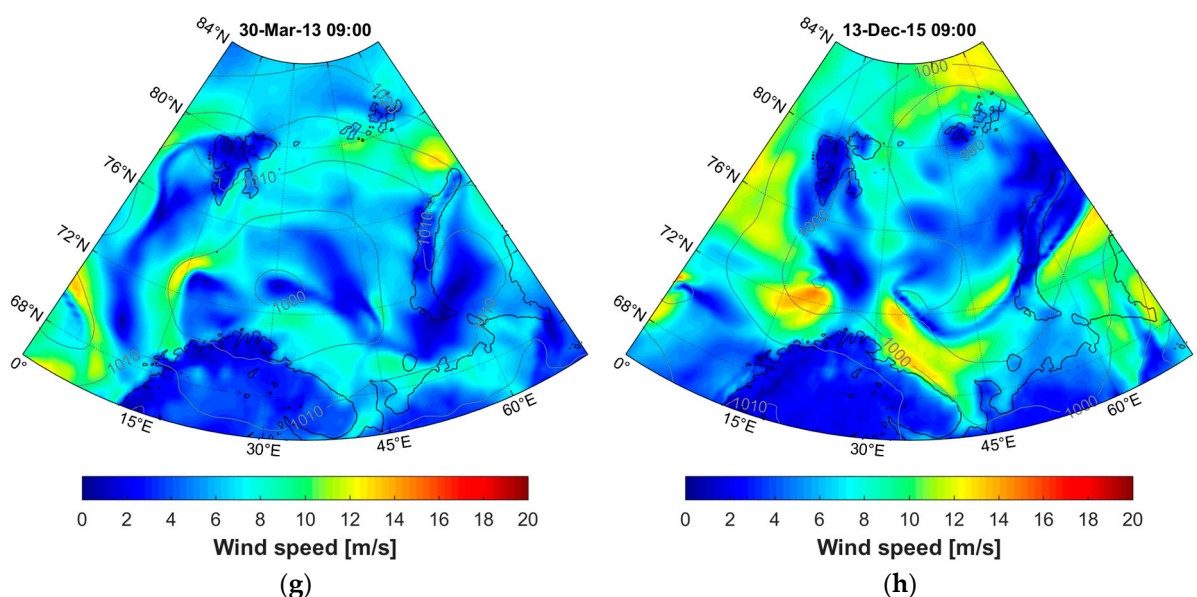

**Figure 7.** Comparison between surface wind speed fields for 30 March, 2013 (**a**,**c**,**e**,**g**), and 13 December, 2015 (**b**,**d**,**f**,**h**), according to AMSR-2 remote sensing data (**a**,**b**), COSMO-CLM hindcast (**c**,**d**), ERA-Interim (**e**,**f**), and ERA5 (**g**,**h**) reanalyses.

The presented cases are only an example of the CCLM dataset application, aimed to show the contrasting atmospheric dynamics via coarser reanalysis data and a high-resolution hindcast. The model's ability to resolve PLs should be systematically addressed in further studies. We hope that the new CCLM dataset may be valuable for studies dealing with the PL climatology, tracking, and case study analyses.

As we aimed to introduce the new Arctic CCLM hindcast to the scientific community and to discuss the first results of its evaluation, there are some limitations to this study. For instance, the huge data volume and its storage issues slow down data processing significantly. Therefore, the presented analysis is limited by major surface fields, and does not cover the whole available period. Our study does not include analysis of the 3D atmospheric fields, which is important to understand the quality of the dataset. A more comprehensive analysis of the dataset beyond the listed limitations is planned for our further studies.

The nearest prospect of the CCLM Russian Arctic dataset application is its comprehensive assessment and comparison with other appropriate datasets, including reanalyses (ERA-5 [22], ASR v.2 [25], NCEP/CFSv2 [23], MERRA2 [21], Arctic CORDEX [29]), gridded archives (HadCRUT4 [89], GPCP [90], etc.) and satellite data (SAR [91], QuikSCAT [92], AMSR-E [93], etc.). This will provide important and useful information regarding the opportunities and restrictions of this dataset in terms of different variables and specific regions to outline the limits of its applicability and obtain a framework of possible future tasks. The next proposed stage of our work will aim to further downscale the dataset to 0.03° grid spacing over three domains, including the Barents, Kara, and Laptev Seas and the corresponding data analysis.

## 5. Conclusions

The COSMO-CLM Russian Arctic hydrometeorological dataset was created using long-term nonhydrostatic regional model simulations for the 1980–2016 period with 0.108° grid spacing. The model configuration was determined in several preliminary test simulations for specific summer and winter periods and in the consequent evaluation of the modeling results against the ground-based observations.

The model setup included a newer model (version 5.05), the ICON-based physical package, and the spectral nudging technique. The use of the recent ERA5 reanalysis data for the model forcing did not demonstrate any noticeable improvements in comparison to the

use of ERAI reanalysis; therefore, the latter was used for the model forcing in the long-term hindcast. A monthly reinitialization scheme with an additional assimilation of the deep soil properties was proposed, tested, and implemented in the long-term simulations.

The dataset creation produced more than 120 Tb total volume and consumed more than 250,000 node-hours during the runs. Therefore, sharing the dataset using any hosting, HTTP, or FTP service is a challenging technical task. In the first stage, we prepared a publicly available subset using the Figshare service (see Section 3.1, Appendix C and [72]) according to the CC BY 4.0 license and citations, which includes seven main surface variables within the entire 37-year period. We plan to consistently extend the list of accessible variables and hope these data will be useful and appropriate for Arctic climate research.

The primary assessments of the surface wind speed and temperature fields showed good agreement with ERA-Interim reanalysis in large-scale patterns and many physically justified added values in the regional mesoscale feature reproduction according to the coastlines, mountains, large lakes, and surface properties. An example of two cases with polar low activity demonstrated a dramatic difference in the specific mesoscale dangerous weather event representation in the developed dataset and in the coarser reanalysis data. The presented dataset is a pioneering example for the Arctic region, combining resolution and time coverage.

The dataset has a wide range of potential applications. It can be used for regional Arctic climate change studies; for a deeper understanding of the physical mechanisms of extreme mesoscale weather events and assessment of their repeatability; investigations of the modern changes in the Russian Arctic environment; as a forcing data source for many detailed extreme event case studies, such as atmospheric forcing for ocean models, including wind waves and full dynamics; in polar low dynamics and climatology studies; in climate resource assessments; and more.

**Author Contributions:** Conceptualization, V.P. and M.V.; methodology, M.V. and V.P.; software, V.P.; validation, M.V.; formal analysis, V.P. and M.V.; investigation, V.P. and M.V.; resources, V.P.; data curation, M.V.; writing—original draft preparation, V.P.; writing—review and editing, M.V. and V.P.; visualization, M.V. and V.P.; supervision, V.P.; project administration, V.P.; funding acquisition, V.P. and M.V. All authors have read and agreed to the published version of the manuscript.

**Funding:** The reported study was funded by the Russian Foundation for Basic Research (RFBR) according to research project No. 18-35-00604 and Lomonosov Moscow State University project no. AAAA-A16-116032810086-4. The APC was funded by RFBR research project No. 18-35-00604. The work of Mikhail Varentsov was partially supported by RFBR project No. 18-05-80065.

**Data Availability Statement:** The data presented in this study are openly available in [FigShare] at [https://doi.org/10.6084/m9.figshare.c.5186714] (accessed on 2 March 2021), reference number [5186714].

**Acknowledgments:** The research was carried out using the equipment of the shared research facilities of the HPC computing resources at Lomonosov Moscow State University.

**Conflicts of Interest:** The authors declare no conflict of interest.

## Appendix A

**Table A1.** List of all test experiments with their properties and acronyms.

| Experiment Acronym | Model Version | Forcing Data | Spin Up | Spectral Nudging | Turbulence Scheme Correction (tkhmin = tkmmin = 0.1) |
|---|---|---|---|---|---|
| COSMO_erai | 5.0 | ERA-Interim | No | No | Standard |
| COSMO_erai_long | 5.0 | ERA-Interim | Yes | No | Standard |
| COSMO_era5 | 5.0 | ERA5 | No | Yes | Standard |
| COSMO_erai_sn | 5.0 | ERA-Interim | No | Yes | Standard |

**Table A1.** *Cont.*

| Experiment Acronym | Model Version | Forcing Data | Spin Up | Spectral Nudging | Turbulence Scheme Correction (tkhmin = tkmmin = 0.1) |
|---|---|---|---|---|---|
| COSMO_era5_sn | 5.0 | ERA5 | No | Yes | Standard |
| COSMO_erai_turb | 5.0 | ERA-Interim | No | No | Corrected |
| COSMO_era5_turb | 5.0 | ERA5 | No | Yes | Corrected |
| COSMO_erai_turb_sn | 5.0 | ERA-Interim | No | Yes | Corrected |
| COSMO_erai_turb_sn_long | 5.0 | ERA-Interim | Yes | Yes | Corrected |
| COSMO_era5_turb_sn | 5.0 | ERA5 | No | Yes | Corrected |
| COSMO_era5_sn_v505 | 5.05 | ERA5 | No | Yes | Standard |
| COSMO_erai_sn_v505 | 5.05 | ERA-Interim | No | Yes | Standard |
| COSMO_erai_v505_long | 5.05 | ERA-Interim | Yes | No | Standard |
| COSMO_erai_sn_v505_long | 5.05 | ERA-Interim | No | No | Standard |

## Appendix B

**Table A2.** List of the main variables included in the COSMO-CLM Russian Arctic hindcast. The data available now in [72] are in **bold**.

| Variable Acronyms | Variable Full Names | Dimensions (2D— Surface/3D—Model Levels) |
|---|---|---|
| U, V, W, T, FI, TKE, POT_VORTIC, H_SNOW, RHO_SNOW, W_SNOW, RELHUM, QV | Zonal, meridional, and vertical velocities, temperature, geopotential, turbulence kinetic energy, Ertel potential vorticity, snow height, density, water content, relative and specific humidity | 3D |
| **U_10M, V_10M,** VMAX_10M, **VABSMX_10M** | Zonal, meridional, maximal velocities, and wind gusts on 10 m | 2D |
| **T_2M,** TMAX_2M, TMIN_2M, TD_2M, TWATER | 2 m temperature, maximal and minimal, 2 m dew point, water temperature | 2D |
| **PMSL,** HPBL | Sea level pressure, planetary boundary layer height | 2D |
| T_S, T_SNOW, T_SO, T_ICE | Surface, snow, soil, ice temperatures | 2D |
| TQC, TQI, TQR, TQS, TQG, TQV, | Vertical integrated cloud water, ice, rain, snow, graupel, precipitable water, total water content | 2D |
| CLCM, CLCH, CLCL, CLCT, CLDEPTH | Medium, high, low, total, convective cloud cover, cloud depth | 2D |
| CLC_CON | Convective cloud area fraction | 3D |
| LHFL_S, SHFL_S | Latent and sensible heat fluxes | 2D |
| SWDIRS_RAD, SWDIFDS_RAD, THDS_RAD, THUS_RAD, SOBS_RAD, THBS_RAD, SWDIFUS_RAD, ALB_RAD | Surface radiation components: shortwave direct and diffuse, longwave downward and upward, net shortwave and longwave radiation, reflected, albedo | 2D |
| RELHUM_2M, **QV_2M** | Relative and specific humidity at 2 m | 2D |
| FRESHSNW, SNOW_MELT | Freshness of snow, snow melt | 2D |
| **TOT_PREC,** SNOW_CON, SNOW_GSP, RAIN_CON, RAIN_GSP, RUNOFF_S, RUNOFF_G | Total precipitation, convective and grid-scale snow, convective and grid-scale rain, surface and subsurface runoff | 2D |
| CAPE_MU, CIN_MU, CAPE_ML, CIN_ML | Convective Available Potential Energy (CAPE) and Convective Inhibition (CIN) indexes of most unstable (MU) parcel, and mean surface layer parcel (ML) | 2D |

**Appendix C**

Free-access public link to the Russian Arctic COSMO-CLM hindcast over the 1980–2016 period: https://figshare.com/collections/Arctic_COSMO-CLM_reanalysis_all_years/5186714 [https://doi.org/10.6084/m9.figshare.c.5186714], accessed on 6 March 2021.

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
