# Peer review of "Introducing a New Detailed Long-Term COSMO-CLM Hindcast for the Russian Arctic and the First Results of Its Evaluation"

_atmosphere, doi:10.3390/atmos12030350_

Round 1
Reviewer 1 Report
The suggestions are described in my review report

Author Response
Responses to Reviewer’s 1 comments on paper
Point. I am very disappointed with the answers of the authors to my previous review.
They introduced simply cosmetic changes in the text and did practically ignore most of my detailed suggestions. The authors' explanations delivered in the answers to my review has to be taken into account in the content of their paper.
If the authors really would like to introduce the Arctic COSMO-CLM to the scientific community, as they stated, than the paper has to be rewritten and enriched with more science, more convincing figures about the added value of the COSMO-CLM simulations in comparison to the reanalysis, all I explained already in my first review.
Response. We have partially taken into account the suggested points and significantly enriched the analysis with seasonal features of the COSMO-CLM hindcast added value to the wind speed and temperature climatology, especially focused on wintertime differences. In particular, we have added the corresponding figures (4d, 5ef) and table (Table 2) with complementary added values statistics accompanying the analysis.
We would like to point out repeatedly that we aimed only to announce the Arctic COSMO-CLM hindcast, without a comprehensive analysis. In our opinion, the announcement of the dataset has to include description of creation technology, first evaluation results, and discussion on potential perspectives of the hindcast. For this reason, we have submitted this manuscript as Communication, which assumed more announcing type of articles and fitted well to our task.
Compared to the first version only two small changes were made by the authors.
Point 1. They changed Fig. 1 but forgot to update the Figure caption, since the removed integration areas are still mentioned. Before resubmission the authors should carefully read their own paper. Such a simple error is embarrassing.
Response 1. The point is fair, there was a mistake due to our oversight. Now we have fixed the caption to “Figure 1. Scheme of the model domain with grid spacing ≈12 km, used for simulations in our study, where its borders are shown by a black rectangle, and the elevation data used in the model are shown.”
Point 2. Fig. 7 contains now additional ERA5 plots for two dates, Why these plots are important for the claimed long-term hindcast simulations was not explained.
Response 2. We have added an additional comparison with more recent and detailed ERA5 data in this case on the request and suggestion by another Reviewer. These panels investigated the ERA5 capability to reproduce given polar lows cases in comparison with the ERA-I dataset and COSMO-CLM hindcast. This demonstrated that as well ERAI, as ERA5 failed to reproduce polar low activity and strongly underestimated the wind speed compared to the COSMO-CLM hindcast. Again, this figure is suggested as a demonstration of the possible applications of the COSMO-CLM hindcast, but not as part of its comprehensive evaluation.
Reviewer 2 Report
Having in mind the author's response, I reckon that much effort has been put in this manuscript.
However I still hesitate accepting some parts (that were not changed). I would not suggest rejection though.
Author Response
Responses to Reviewer’s 2 comments on paper
Having in mind the author's response, I reckon that much effort has been put in this manuscript.
Point. However I still hesitate accepting some parts (that were not changed). I would not suggest rejection though.
Response. Authors are grateful to the Reviewer for recognition of our messages in Responses. However, we have explained in the first Response, why we do not agree with some points, and therefore did not change text in some parts. We would be thankful, if the Reviewer would point out specifically those parts needed to be changed.
Reviewer 3 Report
The work is about the creation of a high-resolution dataset for the Russian Arctic region, using COSMO-CLM model for the period 1980-2016.
The dataset has a big potentiality, because the Arctic is one of the key regions for detecting climate change effects.
General comments
- The main problem is about the language. In my opinion, the paper needs a deep rewriting as far as the English is concerned. In the specific comments below I list only a few examples. The sentences are often not clear, the use of the verbs and of the articles is sometimes odd.
- From a more scientific point of view, I would like to see more statistical analysis of the new dataset with respect to ERA-INTERIM and to observations, not only differences of (few) fields. I am talking about the long period, not the test periods.
- Also more comparisons with satellite data would be beneficial (at the moment there is only fig 7 a,b).
- Have the authors compared the new dataset with the already existing ones (apart from ERA-INTERIM) ?
- I do not like the use of the words “hydrometeorology” /”hydrometeorological” because I do not see any hydrological variable, there are only meteorological/atmospheric ones.
Specific comments
- Line 18: “emphasize” not “emphasizes”
- Line 56: “reach” not “reached”
- Line 57: “reconstruction of” not “reconstruction”
- Line 63: “v1 and v2” not “v1 и v2”
- Line 66: “covering” not “covered”
- Line 70: remove “certainly”
- Model description: how are the lakes treated in CLM? Is the FLAKE parametrisation on?
- Line 131: “allowing to” not “allowing”
- Line 139-140: rephrase
- Fig 1: not clear, I see two rectangles only
- Fig 2, line 206: “show” not “shows”
- Line 225: “was used” not “used”
- Line 232: “are referred”/“are characterized” not “referred”/”characterized”
- Line 280: what is the relaxation zone ? Explain better
- Line 286: “good” not “well”
- Line 292: you mention lakes, but it depends also on the way they are treated in the model (see comment 7)
- Fig 4,5,6,7: the quality does not look good enough. There are also lines appearing between the panels
- Fig 4a,b: the red area in the upper-left corner looks like a spurious effect due to boundary conditions. Are you sure it is a physical effect ?
- Line 321-323 and Fig 5c,d: this is not clear. A better explanation is needed. Why should the difference (COSMO-ERA) be higher in certain areas during the period 1980-1990 than during 2010-2016 ?
- Fig 5b: there is an offset of the panel
- Line 378: “do” not “does”
- Line 379: remove “hover”
- Line 383: “the presented cases are” not “presented cases is”
- Line 386-387: caption of Fig 7, g and h have a different character
- Line 389: “37-years” not “37-year”
- Line 390-391: rephrase
- Line 398: “For instance the huge…” not “The huge…”
- Line 399: remove “which”
- Line 400-402: rephrase
- Line 419: “includes” not “include”
- Line 428-430: rephrase
Author Response
Responses to Reviewer’s 3 comments on paper
General comments
Point 1. The main problem is about the language. In my opinion, the paper needs a deep rewriting as far as the English is concerned. In the specific comments below I list only a few examples. The sentences are often not clear, the use of the verbs and of the articles is sometimes odd.
Response 1. We are very grateful to the Reviewer for pointing out the language issues in specific comments, and take into accounts all specific comments below. The language issues were extensively revised throughout the test by ourselves and then additionally by the MDPI English Editing Service. We have responded to the majority of the specific editing comments provided by Reviewer, point by point, as it was corrected by us or by the editing service. We have also responded to the points concerned to rephrasing or clarifying the text and the sense.
Point 2. From a more scientific point of view, I would like to see more statistical analysis of the new dataset with respect to ERA-INTERIM and to observations, not only differences of (few) fields. I am talking about the long period, not the test periods.
Response 2. We agree with the Reviewer that detailed evaluation of the new data set with respect to observations and reanalysis data is a highly relevant and important task. However, the presented Communication manuscript is aimed only to announce the Arctic COSMO-CLM hindcast, without a comprehensive analysis and evaluation of the data set. The main reason for this was a desire to share this dataset with the scientific community, because the above-mentioned dataset processing could be much more effective in collaboration. We hope that the data set may be valuable for different studies and deserves to be published. We consider the data set presentation to be suitable for the selected Communication format.
Detailed evaluation of the data set against observations and other data sets was not our original aim. Due to the huge data volume, its processing is technically challenging, so it is difficult to perform additional kinds of data analysis in a short time within the framework of paper revision. Nevertheless, we attempted to do our best in enriching the presented analysis as Reviewer suggested. In addition to different plots, we provided the quantitative statistics of the CCLM-ERAI differences for temperature and wind speed to the text and to new Table 2, and analyzed in more details seasonal patterns of such differences for summer and winter. The figures were modified and extended in order to support such analysis by providing new subplots (4d, 5def). More detailed evaluation of the CCLM hindcast against available observations and other gridded datasets will be a primary focus of our further studies. This point is clearly stressed out in the discussion section.
Point 3. Also more comparisons with satellite data would be beneficial (at the moment there is only fig 7 a,b).
Response 3. The Figure 7 is only an example of polar low reproduction compared between the COSMO-CLM hindcast and state-of-the-art global reanalysis data. As we have pointed out in Response 2, our current manuscript is not aimed at comprehensive comparison and evaluation between different data sets. In the presented study, we would like only to introduce the Russian Arctic COSMO-CLM hindcast, to highlight possible applications of the data set (including the applications related to polar low studies) and to outline further research direction. We agree that comprehensive evaluation of our data set, together with other existing ones, against satellite observations is a highly relevant and important task for Arctic climate science, so we plan to follow this idea in our further studies.
Point 4. Have the authors compared the new dataset with the already existing ones (apart from ERA-INTERIM)?
Response 4. As we have mentioned in the Response 2, we did not aim to provide comprehensive comparisons in this manuscript, since this is an announcement of the CCLM Russian Arctic hindcast only. Therefore, there are not any more examples of other datasets evaluation. We have given an example of comparison for the ERA5 reanalysis for two polar lows cases (Figure 7g,h). Comprehensive comparison with other global and regional reanalyses and datasets is planned for our further studies.
Point 5. I do not like the use of the words “hydrometeorology” /”hydrometeorological” because I do not see any hydrological variable, there are only meteorological/atmospheric ones.
Response 5. Authors agree that the main focus of the COSMO-CLM Russian Arctic hindcast is meteorological and climatological. However, one of the hydrometeorological tasks is to study the transfer of water and energy between the land surface and the lower atmosphere. Therefore, our hindcast is proposed as hydrometeorological, since it provides miscellaneous data related to surface water and energy balance evaluations. For instance, precipitation, snow characteristics and properties, net balance and temperature regime are among the most important variables needed for many hydrological and hydroclimatological estimates including extremes and risks assessment. Since the COSMO-CLM Russian Arctic hindcast allows us to study various hydrological problems, we suppose that it could be referred to as a hydrometeorological dataset.
Specific comments
Point 1. Line 18: “emphasize” not “emphasizes”
Response 1. Changed from “emphasizes” to “emphasize”, line 15.
Point 2. Line 56: “reach” not “reached”
Response 2. Changed from “reached” to “reach”, line 57.
Point 3. Line 57: “reconstruction of” not “reconstruction”
Response 3. Changed from “reconstruction” to “reconstruction of”, line 57.
Point 4. Line 63: “v1 and v2” not “v1 и v2”
Response 4. Changed from “v1 и v2” to “v1 and v2”, line 65.
Point 5. Line 66: “covering” not “covered”
Response 5. Changed from “covered” to “covering”, line 69.
Point 6. Line 70: remove “certainly”
Response 6. The word “certainly” removed, line 71.
Point 7. Model description: how are the lakes treated in CLM? Is the FLAKE parametrisation on?
Response 7. Yes, the FLAKE parameterization was on, and lakes parameterized in the COSMO-CLM model, including the cold start option. The explanation text with reference was added in lines 122-123.
Point 8. Line 131: “allowing to” not “allowing”
Response 8. Changed from “allowing” to “allowing to”, line 144.
Point 9. Line 139-140: rephrase
Response 9. The sentence rephrased as: “This binds the large-scale components of the internal model mode to the forcing data, and prevents the model from moving away from realistic large-scale patterns of atmospheric circulation.”, lines 152-154.
Point 10. Fig 1: not clear, I see two rectangles only
Response 10. It was fixed, the caption was changed to: “Figure 1. Scheme of the model domain with grid spacing ≈12 km, used for simulations in our study, where its borders are shown by a black rectangle, and the elevation data used in the model are shown.”, lines 187-188.
Point 11. Fig 2, line 206: “show” not “shows”
Response 11. Changed from “shows” to “show”, line 227.
Point 12. Line 225: “was used” not “used”
Response 12. Changed “used” to “was used”, line 247.
Point 13. Line 232: “are referred”/“are characterized” not “referred”/”characterized”
Response 13. Changed from “referred”/”characterized” to “are referred”/“are characterized”, lines 255-256.
Point 14. Line 280: what is the relaxation zone? Explain better
Response 14. The explanation of the relaxation zone with references added in lines 312-314.
Point 15. Line 286: “good” not “well”
Response 15. Changed from “well” to “good”, line 320.
Point 16. Line 292: you mention lakes, but it depends also on the way they are treated in the model (see comment 7)
Response 16. The lakes are treated with the FLAKE parameterization, as responded to at point 7.
Point 17. Fig 4,5,6,7: the quality does not look good enough. There are also lines appearing between the panels
Response 17. The figures were changed by better quality at the previous stage of review. The point is not so clear according to lines between the panels.
Point 18. Fig 4a,b: the red area in the upper-left corner looks like a spurious effect due to boundary conditions. Are you sure it is a physical effect?
Response 18. If the Reviewer mentioned the red area over the northern Atlantic, there is not the spurious effect due to boundary conditions, because it is manifested in the global driving conditions, ERA-I data, which does not have boundaries in this area. Therefore, the corresponding red area in the COSMO-CLM data is a physical effect and detailed wind speed frequency pattern according to global forcing.
Point 19. Line 321-323 and Fig 5c,d: this is not clear. A better explanation is needed. Why should the difference (COSMO-ERA) be higher in certain areas during the period 1980-1990 than during 2010-2016?
Response 19. The mentioned text rephrased, and more details added. Moreover, two panels added with seasonal patterns of the CCLM-ERAI differences, and analyzed in text (lines 363-384)
Point 20. Fig 5b: there is an offset of the panel
Response 20. An offset of the panel of Fig. 5b was fixed.
Point 21. Line 378: “do” not “does”
Response 21. This text was rephrased by Editing Service without “do”.
Point 22. Line 379: remove “hover”
Response 23. The word “hover” removed, line 400.
Point 24. Line 383: “the presented cases are” not “presented cases is”
Response 24. Changed from “presented cases is” to “the presented cases are”, line 460.
Point 25. Line 386-387: caption of Fig 7, g and h have a different character
Response 25. We did not understand this point. From our point of view, all captions are correct.
Point 26. Line 389: “37-years” not “37-year”
Response 26. The English Editing Service has written this as “37-year”.
Point 27. Line 390-391: rephrase
Response 27. The sentence rephrased as: “Therefore, its sharing using any hosting, HTTP or FTP service is a challenging technical task.”, lines 499-500.
Point 28. Line 398: “For instance the huge…” not “The huge…”
Response 28. Changed from “The huge…” to “For instance, the huge…”, lines 468-469.
Point 29. Line 399: remove “which”
Response 29. The word “which” removed, line 422.
Point 30. Line 400-402: rephrase
Response 30. The sentence rephrased as: “Therefore, the presented analysis is limited by major surface fields, and does not cover the whole available period. Our study does not include analysis of the 3D atmospheric fields, which is important to understand the quality of the dataset. A more comprehensive analysis of the data set beyond the listed limitations is planned for our further studies.”, lines 470-474.
Point 31. Line 419: “includes” not “include”
Response 31. Changed from “include” to “includes”, line 502.
Point 32. Line 428-430: rephrase
Response 32. The sentence rephrased as: “An example of two cases with polar lows activity demonstrated a dramatic difference in the specific mesoscale dangerous weather event representation in the developed dataset and in coarser reanalysis data.”, lines 508-511.
Round 2
Reviewer 1 Report
The english needs polishing by a native speaker.
Author Response
The manuscript was double checked by the MDPI English Editing Service. Therefore, we believe that English is satisfactory in our manusript.
Reviewer 2 Report
I would suggest two major changes.
First I would suggest to exclude everything about the ERA5 dataset because it is misleading.
Secondly I would suggest to evaluate (against observations) the wind and temperature extremes (statistics, QQplots etc) in order to show the added value of the regional modeling against the EI dataset.
Author Response
Responses to points by Reviewer 2.
General comment: We are thankful to the Reviewer for reading our manuscript and providing fruitful ideas of its improvements. However, we would like to highlight the given Article is a Communication Paper, that is aimed just to introduce the developed Arctic COSMO-CLM hindcast to the scientific community without a comprehensive analysis aimed at evaluation of the whole data set. Main motivation for our decision is a desire to share the data set that was born after long-term and expensive computations, to the scientific community, and to give opportunity for its analysis by different groups and researchers, following the modern principles of FAIR (Findability, Accessibility, Interoperability, Reusability) scientific data management.
We agree with Reviewer that there are numerous types of data analysis that are essential for the new data set, including its comprehensive statistical evaluation and comparisons with other data sets. However, almost all types of the data analysis and evaluation that are foreseen, in our opinion deserve accurate and detailed explanation and publication as separate studies, but not as short and limited part of this study.
Presented manuscript in our opinion is detailed enough and saturated for the Communication paper, announcing the new data set. In addition to the technical details on the model configuration, computations, and data set structure, we present analysis of its added value to the temperature and wind climatology, and discuss the possible scenarios of its application.
In our further studies we plan to continue evaluation of the data set and estimate the COSMO-CLM Arctic hindcast based on other detailed data sources including reanalyses, datasets, satellite observations, etc. including extreme statistics evaluation. The COSMO-CLM Arctic hindcast evaluation and comparison with ERA5 and ASRv2 data is one of the important goals undoubtedly, which we hope to fulfill and publish in the next papers. Additionally, we hope that when the data set will become announced and publicly available, it will attract interest of other research groups, allowing engaging them to further data analysis.
We hope the given explanations argue the aims and scope of this manuscript as Communication. Therefore, we give Responses further taking into account the above-mentioned clarifications.
Point 1. First I would suggest to exclude everything about the ERA5 dataset because it is misleading.
Response 1. We have included additional figure panels and corresponding discussion concerning the ERA5 reproduction of two polar lows cases according to suggestion Reviewer 2 (i.e., by you) at the first stage of revision (Point 5. "In lines 356-366 the authors show that EI fails to reproduce polar lows and the extremes associated with them. This leads to some questions. 1. Have you checked the performance of E5 as well in this case? It would be interesting to demonstrate."). Therefore, we are a bit confused with a suggestion to exclude all about the ERA5 dataset. In our opinion, Figure 7g,h presented an important example demonstrating the performance of the COSMO-CLM hindcast in comparison not only with ERAI data used as model forcing, but with more recent and detailed ERA5 dataset. These panels indicate that ERA5 reanalysis with ~30 km grid size failed to reproduce the PL activity for the first case, and strongly underestimated the corresponding wind extremes in the second case. It is important because the presented cases are an example of the CCLM dataset application, aimed to show the contrasting atmospheric dynamics via coarser reanalysis data and a high-resolution hindcast.
Point 2. Secondly I would suggest to evaluate (against observations) the wind and temperature extremes (statistics, QQplots etc) in order to show the added value of the regional modeling against the EI dataset.
Response 2. We agree that model-to-observation comparison for wind and temperature extremes is essential for the evaluation of the developed data set, and qq-plots are a nice visualization tool to present such results. However, as we highlighted above, we would like to avoid including comprehensive evaluation of the data set against observations in current manuscript. In our opinion, such analysis deserves to be carefully performed and published as a separate paper obviously. Observation network in the Arctic is scarce and inhomogeneous, stations may be located in different local-scale settings (at the coast, or in the mountains), time series may include significant gaps or errors, that should be checked and filtered out. All these issues should be accurately taken into consideration and well-described when presenting the results. However, a detailed description in our opinion is far beyond the scope of our current study. Additionally, the data for many Russian Arctic weather stations has restricted access, so additional work and time is needed to get these data. Therefore, we do not see any reasonable opportunity to include comprehensive model-to-observation comparison for extremes in current paper, due to significant time demands for performing such analysis, and due to the limitations of the Communication paper format to present it with enough details. Considering these limitations, it is possible to present only a brief example of such model-to-observation comparison for temperature and wind extremes for one or two sites. However, we would like to avoid doing this, since such examples may be biased and unrepresentative due to the specific features of selected stations. Model-to-observation comparison for the Russian Arctic CCLM hindcast and other global data sets for weather extremes will be the primary focus of our further studies.
This manuscript is a resubmission of an earlier submission. The following is a list of the peer review reports and author responses from that submission.
Round 1
Reviewer 1 Report
see the detailed review as pdf submitted to the journal as a separate file

Author Response
Responses to Reviewer’s 1 comments on paper
Authors are very grateful to the Reviewer for thorough and critical reading of our Article. Many of the issues pointed by Reviewer are fair and deserved. However, we would like to highlight that the given Article is a Communication Paper, that is aimed just to introduce the developed Arctic COSMO-CLM hindcast to the scientific community. Reviewer is almost right in points concerned to more careful and qualified verification, statistical evaluation, comparisons, and assessment of this dataset. All of this is planned to be fulfilled in the further studies by the Authors and published as separate papers.
In presented Communication manuscript, we would like only to announce the Arctic COSMO-CLM hindcast, without a comprehensive analysis aimed at evaluation of the data set. There were some reasons for this choice. The first one was a desire to share this dataset with the scientific community, because the above mentioned dataset processing could be much more effective in collaboration. Despite the many shortcomings of the data set indicated by Reviewer, up to our knowledge this is the only meteorological dataset with so long temporal coverage (37 years) and so high spatial resolution (~13 km). This gives us a hope that the data set may be valuable for different studies and deserves to be published.
The second reason is the huge data volume, which could slow down the data processing significantly. Therefore, we have focused on the main surface fields analysis, excluding detailed assessment of 3D fields, which is also very important to understand the quality of the dataset. The third one was the scope of Communication paper format, which assumed more announcing type of articles and fitted well to our task, because we could not include the whole period and many variables in the analysis. Besides, much more detailed information about preliminary experiments and their evaluation was referred to [Platonov, V., Varentsov, M., 2019, https://doi.org/10.1088/1755-1315/386/1/012039]. We did not include all this information in the current paper, because these questions were not in the focus (although, these questions are important, of course).
In Authors’ opinion, the given reasons argued and narrowed aims and scope of this paper, which does not cancel the fair criticism and rightness of the Reviewer in many general issues. We are very grateful for many points and tried to address them during revision of this paper as well as in further studies. Authors gave all Responses further taking into account the above mentioned clarifications.
Responses to Reviewer 1 Comments
Point 1. The current model setup leaves many questions open and must be improved. The authors should establish contacts to groups with more experience in evaluating regional model simulations in the Arctic, e. g. ARCTIC-Cordex and apply some of their analysis tools, e. g. Akperov et al. JGR 2018 or Sedlar et al. 2020 cited at the end of this review.
Response 1. We are grateful to the Reviewer for pointing out information about Arctic CORDEX experiments. When configuring the model set up, we contacted S. Kohnemann and G. Heinemann from University of Trier, which evaluated COSMO-CLM runs within the Arctic CORDEX project and used their recommendations to configure the model. have also contacted M. Akperov and planned the next common works on evaluation and comparison of Arctic CORDEX results with COSMO-CLM Arctic hindcast, since there is a large separate task. Nevertheless, the simulations have been already performed using the presented model set up, and millions of CPU hours have been already spent for computations. Since the developed data set is the only one for the Arctic that combines so high resolution and temporal coverage, we hope that the resources are well spent, and the data set, once published, will be useful for the scientific community.
Point 2. The described experiments have to be evaluated in a more qualified way and more exactly with a focus on scientific questions, which are missing in the current version.
Response 2. As we already highlighted above, the scientific question of our study is the development of the COSMO-CLM Arctic hindcast, and the manuscript is aimed to present this data set to the scientific community, but not to its detailed evaluation.
Point 3. The comparison of boundary forcing itself (ERA-I or ERA-5) needs more than the simple Table 1 with domain averaged quantities and does not allow any quality check. You have to compare these results much more carefully.
Response 3. We agree with the Reviewer that this part of the study may be described in much more detail and even be published as a separate study. And of course, more detailed comparison was performed when making decisions about the forcing data selection. Nevertheless, we could not agree that results from Table 1 are unconvincing. For all considered pairs of the experiments (with and without spectral nudging, with old and new model versions) the use of ERA5 forcing does not provide any clear advantage over ERAI. Taking into account much higher storage and processing time for ERA5 data, we consider the ERA-Interim forcing to be a cost-effective solution for the long-term simulations. We attempted to explain this issue in a more clear way in the revised manuscript (see lines 191 - 195).
Point 4. The described preliminary numerical experiments are not sufficiently evaluated and do not meet the required scientific quality standards.
Response 4. We have used the most common and widespread statistics for evaluation including bias, RMSE, correlation coefficients. We would be thankful for a more detailed explanation, what specific quality standards are not met in the presented description.
Point 5. The whole model setup 2.2.1 has to be rewritten. The selection of a best model configuration leaves many questions open. The vertical resolution should be mentioned. The additional domains in Fig. 1 should be removed and the authors should focus on the evaluation of their simulations in the basic domain. The smaller domains would require a separate evaluation. Table A2 is not needed.
Response 5. Information about vertical resolution was added in text, domains on the Figure 1 were fixed. We consider Table A2 as an important information source containing the whole list of variables, potentially available in the COSMO-CLM Arctic hindcast. We suppose to leave this Table and information, because it is useful and interesting for many future dataset users and could serve as a basis for any possible collaborations.
Point 6. You apply ERA-I lateral forcing data with 0.75° and run COSMO-CLM with 12 km. Would this setup not require a double nesting?
Response 6. We used this downscaling approach (from 0.75° to 0.108°) quite surely, because there are many reliable examples of application of this scheme in COSMO-CLM simulations [M. Tolle, 2020, https://doi.org/10.3390/su12156140; Zentek, Heinemann, 2020, https://doi.org/10.5194/gmd-13-1809-2020]. Besides, according to CLM-Community recommendations for boundary conditions, downscaling factors higher than 17 are not recommended [Pavlik et al., 2012, https://doi.org/10.1007/s12665-011-1081-1]. In our case the downscaling factor is approximately equal to 6, which fits well to these recommendations. It is also necessary to take into account that the real grid size of ERA-Interim data in the Arctic is smaller than ~75 km, because the 1° distance (in km) is decreasing rapidly in high latitudes.
Point 7. The impact of applied spectral nudging has to be shown in more detail. Why wave scales above 500 km are assimilated?
Response 7. We agree with Reviewer that selection of spectral nudging parameters is always some kind of tradeoff. Scale of 500 km was selected as a lower boundary of the synoptic scale of atmospheric processes in order to bind a large-scale circulation to the reanalysis data, but to avoid disturbing smaller-scale processes. There are many papers indicating an optimal range of the cut-off value of wave length in the spectral nudging approach, and the decision depends on the research tasks significantly. For example, [Da Silva, Camargo, 2018, https://doi.org/10.3389/feart.2018.00232] showed that nudging cases applied with higher wave numbers, corresponded to smaller lengths (up to 500 km), improve the overall performance of simulations. However, [Gomez, Miguez-Macho, 2017, https://doi.org/10.1002/qj.3032] indicated the model error diminishes rapidly as the nudging expands over broader parts of the spectrum, but this decreasing trend ceases sharply at cut‐off wave numbers equivalent to a length‐scale of about 1000 km, and the error magnitude changes minimally thereafter. Some other investigations [Von Storch, et al., 2000, https://doi.org/10.1175/1520-0493(2000)128%3C3664:ASNTFD%3E2.0.CO;2; Feser, Barcikowska, 2012, https://doi.org/10.1088/1748-9326/7/1/014024; Miguez-Macho, G. et al., 2004, https://doi.org/10.1029/2003JD004495; Radu et al., 2008, https://doi.org/10.1111/j.1600-0870.2008.00341.x] demonstrated the 500 km is a sufficient threshold for assimilation wave scales from global driving conditions for the domains of similar scale. In this way, the 500 - 1000 km is obviously the reasonable range of the cut-off wave length value. Besides, Authors’ experience included different long-term COSMO-CLM runs with the spectral nudging [Kislov et al., 2018 https://doi.org/10.1134/S0001433818040242; Varentsov et al., 2018, http://dx.doi.org/10.3390/atmos9020050], used the same cut-off and the modelling results improvement was consistent with the above cited papers in general. A sentence justifying the selected scale was added to the revised manuscript (see lines 141 - 145).
Point 8. The stable turbulence parameterization needs also much more careful evaluation.
Response 8. We agree that representation of the stable atmospheric boundary layers in the models remains a challenging scientific task and generally deserves much more detailed investigation. However, our study is not about this important, but narrow problem. We would like just to demonstrate that we have tested different possibilities to mitigate the known model biases. The solution, selected for the final model runs (new model version with ICON-based physical package) is based on the experience of the model developers, who suggested the revised turbulence scheme to mitigate known problems of the older model versions. Moreover, the positive effect from the use of this solution is confirmed by other cites studies as well by the verification scores, presented in the study. We agree that alternative solution with older model version (*_turb runs) may raise more questions, however it is not used for the final model runs, so we do not see a need to give more attention to the indicated problem in our study. Nevertheless, we attempted to provide a more clear explanation of this problem in the revised manuscript (see lines 147 - 154).
Point 9. Very disappointing is also the handling of soil properties and the arguments to justify the reinitialisation merging deep soil variables from ERA-I with COSMO-CLM simulation results. Such an approach is very questionable!
Response 9. We agree that the supposed scheme is quite questionable and discussionable. However, the conducted test runs and its evaluation have shown absence of any statistically significant changes and errors by introduction of this scheme. We plan to continue investigation of this problem in further studies involving the available soil moisture observations.
Point 10. In the result section the claimed simulation 1980-2016 were not described and evaluated, instead two arbitrary selected time periods 1980-1990 and 2010-2016 are compared for surface wind speed and temperature climatology and extremes. E. g. the statement at LN 302-305 is insufficient.
Response 10. Since the scope of this paper did not aim to embrace the whole period and all variables assessment and taking into account issues and restrictions of data volume, we have presented here two different periods only, and its preliminary comparison.
Point 11. Again the authors have to be much more careful in validating their results. I am also missing critical self reflection since the whole manuscript delivers the impression that it was collapsed in a big hurry. Also any discussion about physical mechanisms behind the changes between the two periods is missing.
Response 11. We claimed that the presented paper does not pretend to be a full description, evaluation, and detailed physical analysis of results. The main goal was to present and announce a new Arctic long-term hindcast, with a brief description of each stage of its creation, verification, and preliminary assessments presentation.
Point 12. The authors introduction is too long and general with many redundant citations. The title is not reflecting the described work. Unfortunately the authors are not aware and did not cite recent important papers on regional climate simulations, e. g.:
Sedlar, J., et al., 2020, Confronting Arctic troposphere, clouds, and surface energy budget representations in regional climate models with observations. J. Geophys. Res. 124, doi.org/10.1029/2019JD031783.
Heinemann, G., 2020, Assessment of regional climate model simulations of the katabatic boundary layer structure over Greenland. Atmosphere, 11, 571, doi:10.3390/atmos11060571
Zhou X., et al., 2019, Simulating Arctic 2-m air temperature and its linear trends using the HIRHAM5 regional climate model, Atmospheric Research 217, 137-149, DOI: 10.1016/j.atmosres.2018.10.022
Akperov, M., et al., 2018, Cyclone activity in the Arctic from an ensemble of regional climate models (Arctic CORDEX). J. Geophys. Res. Atmos. https://doi.org/10.1002/2017JD027703.
Response 12. We have revised the Introduction section, focusing more on the paper’s goal and tasks, updated the title, deleted some redundant references, and have included the suggested references. Authors do not agree that the title is not reflecting the described work, because the work is presenting the preliminary results and announcing a new Arctic hindcast without the whole period analysis. We find the Article’s title is fitting well to the declared goal, tasks, and main contents of the paper. We have added the word “Introducing” in the beginning of the title to emphasize the corresponding goal of the paper.
Reviewer 2 Report
This is a quite interesting manuscript about the first results and the evaluation of a new detailed long-term COSMO-CLM hindcast for 2 Russian Arctic. The manuscript is generally well structured and well written in terms of English use while the main objectives are clear. In order however this manuscript to be accepted for publication some issues have to be addressed.
Minor Remarks
- Lines 50-51 need to be rephrased.
- Figures 3 to 6 are blurry. Please upload figures of better quality.
Major Remarks
- The authors decided to use Era Interim (EI) as input for the long term runs. Could you be more specific on the reasons you chose that over Era5 (E5)? E5 seems to perform really well.
- In the section "3.2. Extreme wind speed and temperature estimates" the authors compare the Cosmo output against the EI output. I wonder if this is a fair comparison since EI is used for boundary and lateral conditions.
- In lines 356-366 the authors show that EI fail to reproduce polar lows and the extremes associated with them. This leads to some questions.
- Have you checked the performance of E5 as well in this case? It would be interesting to demonstrate.
- Since EI does not represent extreme winds well enough, why use it for the comparison in extremes, in section 3.2?
- If E5 is found to better represent the extremes as compared against the EI dataset, why not use them for input to the regional model.
Author Response
Responses to Reviewer 2 Comments
Authors are grateful to the Reviewer for reading the article and generally positive estimations. Below are the responses on the main Reviewer’s points.
Point 1. Lines 50-51 need to be rephrased.
Response 1. Text was rephrased.
Point 2. Figures 3 to 6 are blurry. Please upload figures of better quality.
Response 2. Figures 3 - 6 were substituted by the higher resolution ones.
Point 3. The authors decided to use Era Interim (EI) as input for the long term runs. Could you be more specific on the reasons you chose that over Era5 (E5)? E5 seems to perform really well.
Response 3. We agree that ERA5 is the last generation reanalysis and it is in general characteristics better than any previous ones. However, in this work, in part of experiments design and its evaluation, we have not analyzed and compared ERA-I and ERA5 itself but analyzed the effects from the used ERA5 or ERAI as a model forcing on the regional COSMO-CLM results. Evidently, we have also expected an advantage of ERA5, but evaluation results, generally shown in Table 1 or more detailed in [Platonov, V., Varentsov, M., 2019, https://doi.org/10.1088/1755-1315/386/1/012039], displayed an absence of any statistically significant differences between the sets of experiments with ERA-I and ERA5 forcing. Some differences in statistics (biases, correlations, RMSE) were of the order of 0.1, and of opposite signs. Therefore, we have not revealed any certain advantages of ERA5 forcing. Taking into account much higher storage and processing time for ERA5 data, we consider the ERA-Interim forcing to be a cost-effective solution for the long-term simulations. We attempted to explain this issue in a more clear way in the revised manuscript (see lines 191 - 195).
Point 4. In the section "3.2. Extreme wind speed and temperature estimates" the authors compare the Cosmo output against the EI output. I wonder if this is a fair comparison since EI is used for boundary and lateral conditions.
Response 4. We consider this ERA-I - COSMO-CLM comparison as a logical first step to assess and verify the quality of the COSMO-CLM Arctic hindcast. If we compare COSMO-CLM dataset with its forcing dataset (ERA-I), we could indicate added value, or significant changes in climatology patterns (surface wind speed and temperature, in this case) defined by regional model, its dynamics and more detailed surface properties taken into account [Paeth H., Mannig B., 2013, https://doi.org/10.1007/s00382-012-1517-7; Lenz C.J., et al. 2017, https://doi.org/10.1007/s00382-017-3562-8; Souverijns N. et al., 2019, https://doi.org/10.1029/2018JD028862; Kelemen F. D. et al., 2019, https://doi.org/10.3390/atmos10090537]. Of course, the presented comparison of COSMO-CLM and ERA-I is not the final stage of the hindcast evaluation. The first estimates verified the satisfactory quality of the analyzed dataset and showed some regional details of surface wind speed and temperature patterns. In the next papers and approaches we would compare and estimate the COSMO-CLM Arctic hindcast based on other detailed data sources including reanalyses, datasets, satellite observations, etc.
Point 5. In lines 356-366 the authors show that EI fail to reproduce polar lows and the extremes associated with them. This leads to some questions. 1. Have you checked the performance of E5 as well in this case? It would be interesting to demonstrate.
Response 5. We are thankful to the Reviewer for the suggestion. Comparison with ERA5 data was added. ERA5 reanalysis also failed to reproduce polar low activity for the first case. For the second case, ERA5 reanalysis reproduced the location of the polar lows, but strongly underestimated the wind speed in these areas. (Lines 380 - 382 added)
Point 6. Since EI does not represent extreme winds well enough, why use it for the comparison in extremes, in section 3.2?
Response 6. Comparing the COSMO-CLM Arctic hindcast with the ERA-I reanalysis, we aimed to demonstrate principal advantages of a new dataset including the extremes evaluation. This comparison, also referred to as the added value of the regional modelling, revealed many significant differences in extreme reproduction, which are described in text in detail.
Point 7. If E5 is found to better represent the extremes as compared against the EI dataset, why not use them for input to the regional model.
Response 7. See the Response 3 for reasons to ERA-I choice as driving conditions. The above mentioned evaluation of test experiments was concerned with overall statistics without any specific focus on extreme values. This was done to validate model capability to reproduce main characteristics well. Extreme values analysis is the special objective applied to the final hindcast version, not to the test experiments. Timeseries in test experiments were too short to get robust extreme statistics. The COSMO-CLM Arctic hindcast evaluation and comparison with ERA5 data is one of the important goals undoubtedly, which we hope to fulfill and publish in the next papers.